# The structure of tyrosine-10 favors ionic conductance of Alzheimer's disease-associated full-length amyloid-β channels

Abhijith G. Karkisaval [1], Rowan Hassan[2], Andrew Nguyen[3], Benjamin Balster[3], Faisal Abedin [2,4], Ratnesh Lal [1,3] ✉ & Suren A. Tatulian [2] ✉

Amyloid β (Aβ) ion channels destabilize cellular ionic homeostasis, which contributes to neurotoxicity in Alzheimer's disease. The relative roles of various Aβ isoforms are poorly understood. We use bilayer electrophysiology, AFM imaging, circular dichroism, FTIR and fluorescence spectroscopy to characterize channel activities of four most prevalent Aβ peptides, $Aβ_{1-42}$, $Aβ_{1-40}$, and their pyroglutamylated forms ($AβpE_{3-42}$, $AβpE_{3-40}$) and correlate them with the peptides' structural features. Solvent-induced fluorescence splitting of tyrosine-10 is discovered and used to assess the sequestration from the solvent and membrane insertion. $Aβ_{1-42}$ effectively embeds in lipid membranes, contains large fraction of β-sheet in a β-barrel-like structure, forms multi-subunit pores in membranes, and displays well-defined ion channel features. In contrast, the other peptides are partially solvent-exposed, contain minimal β-sheet structure, form less-ordered assemblies, and produce irregular ionic currents. These findings illuminate the structural basis of Aβ neurotoxicity through membrane permeabilization and may help develop therapies that target Aβ-membrane interactions.

Aberrant fibrillar deposits of the amyloid β (Aβ) peptide constitute a major histopathological trait of Alzheimer's disease (AD) brains[1,2]. There is compelling evidence that the soluble oligomers of Aβ exert the main neurotoxic effect[3-5]. Aβ occurs in different isoforms; while the 42- and 40-residue peptides ($Aβ_{1-42}$ and $Aβ_{1-40}$) are the most abundant species, shorter forms resulting from proteolysis are also present[6]. Of particular interest are the N-terminally truncated and pyroglutamylated Aβ peptides (AβpE) as they are present in AD brains in substantial amounts (up to 25% of total Aβ) and are hypertoxic[7-10]. The augmented cytotoxicity of AβpE has been attributed to elevated β-sheet formation and fibrillogenesis propensity[7,11-13]. However, other publications have shown similar β-sheet structure and slower fibril formation by AβpE species as compared to their unmodified counterparts[14-17].

The molecular mechanisms of Aβ neurotoxicity is complex and not fully understood. Aβ can damage neurons through dysregulation of cellular ionic homeostasis by two main routes: (i) modulation of certain ion channels in plasma or intracellular membranes[18,19] or (ii) permeabilization of cell membranes through direct membrane insertion and ion channel formation[7,20-33]. $Aβ_{1-42}$ and $Aβ_{1-40}$ induced ion conductance in lipid membranes[20,27,34]. Patch clamp experiments and optical imaging of ion channel conductance on neurons identified cation-selective, $Zn^{2+}$-sensitive channels induced by $Aβ_{1-42}$ and $Aβ_{1-40}$, much like those observed in lipid bilayers[21,22,27,35]. Significantly, channel activity of $Aβ_{1-42}$ was stronger in the membranes prepared using brain extract lipids[28]. Moreover, these effects have been eliminated by peptides or small molecules that specifically block Aβ channels[5,23-25], inhibit Aβ aggregation and membrane insertion[29], or competitively prevent the binding of Aβ to membrane lipids[36]. In HEK293 cells, $Aβ_{1-42}$ oligomers formed ion channels but $Aβ_{1-40}$ did not show channel activity in any aggregation state[30].

[1]Department of Mechanical and Aerospace Engineering, University of California San Diego, La Jolla, CA, USA. [2]Department of Physics, University of Central Florida, Orlando, FL, USA. [3]Department of Bioengineering, University of California San Diego, La Jolla, CA, USA. [4]Present address: Department of Biology, Xavier University of Louisiana, New Orleans, LA, USA. ✉e-mail: rlal@ucsd.edu; statulia@ucf.edu

The inability of $A\beta_{1-40}$ to form channels in cell membranes echoes with the absence of electrical activity of lipid membranes upon addition of micelle-solubilized $A\beta_{1-40}$, in contrast to $A\beta_{1-42}$[37], or non-specific membrane perturbation by $A\beta_{1-40}$ through binding to the bilayer surface[38].

Membrane pore formation and membranotropic neurotoxicity of $A\beta pE$ has been studied less extensively. $A\beta pE_{3-42}$ oligomers were shown to insert into anionic lipid membranes and form $Zn^{2+}$-sensitive channels more efficiently than $A\beta_{1-42}$[39,40]. $A\beta pE_{3-42}$ also allowed higher $Ca^{2+}$ influx into cultured mouse cortical neurons than the unmodified $A\beta_{1-42}$ peptide[7]. Membrane channel data for $A\beta pE_{3-40}$ have not been reported to the best of our knowledge.

Structures of $A\beta$ peptides have been examined by computational and experimental methods. $A\beta_{1-40}$ channels have been simulated based on secondary structure predictions as annular hexameric or dodecameric assemblies where the pore is lined by the polar face of an N-terminal amphipathic β-hairpin while the C-terminal hydrophobic α-helix is in contact with membrane lipids[41,42]. Atomic force microscopy (AFM) images of lipid membranes with reconstituted $A\beta_{1-42}$ identified channel-like structures composed of four or six subunits with an overall outer diameter of 8–12 nm[20]. Meanwhile, spectroscopic studies revealed β-sheet-rich secondary structure for membrane-embedded $A\beta_{1-40}$ and $A\beta_{1-42}$[38,43]. Based on this information and NMR-derived structures of $A\beta_{1-40}$ and $A\beta_{1-42}$ fibrils[44–46], molecular dynamics (MD) simulations produced structural models for channel-forming $A\beta$ fragments N9 and p3 where 12−36 monomers form annular structures composed of loosely connected subunits, each containing 3-7 H-bonded U-shaped strand-turn-strand motifs with N- and C-terminal strands oriented inward and outward, respectively[4,47,48]. Similar structures of $A\beta_{1-42}$ were modeled where six antiparallel β-barrels assemble to form a hexamer of hexamers or rearrange into one annular structure providing a transmembrane pore lined by the N-terminal polar residues[49]. Another $A\beta_{1-42}$ pore model was proposed as a tetramer or hexamer of transmembrane 2-stranded β-sheets upstream of residue 17 with N-terminal β-hairpins extending into the aqueous phase[50]. Further MD studies generated a β-barrel pore model for $A\beta pE_{3-42}$ that comprised 18 tilted transmembrane U-shaped structures (residues 11-15 to 42) with an unordered N-terminus tending to insert into the membrane due to the apolar terminal lactam ring[39]. The β-barrel or β-sandwich-like structure was confirmed for $A\beta_{1-42}$ in detergent micelles and detergent/lipid bicelles by solution NMR[37,51]. More details on the structural aspects of channel/pore formation by various $A\beta$ species can be found in a recent review by Viles[32].

The above discussion underscores the paucity or absence of experimentally determined molecular structures of membrane-embedded $A\beta$ species that play central role in AD etiology, i.e., $A\beta_{1-42}$, $A\beta_{1-40}$, $A\beta pE_{3-42}$, $A\beta pE_{3-40}$. Combined with inconsistent data or absence of data on ion channel formation by some important forms of $A\beta$, e.g., $A\beta_{1-40}$ and $A\beta pE_{3-40}$, it is imperative to undertake a systematic analysis of channel formation abilities and structures of these peptides reconstituted in lipid bilayers. Here we report previously uncharacterized relationship between the molecular structure and morphology of four $A\beta$ isoforms, namely, $A\beta_{1-42}$, $A\beta_{1-40}$, $A\beta pE_{3-42}$, $A\beta pE_{3-40}$ and their ion channel activities. $A\beta_{1-42}$ efficiently embeds in membranes, displays maximum β-sheet structure, forms multisubunit annular assemblies in membranes, and induces step-like single channels. The other three peptides display less effective membrane insertion, smaller β-sheet content, form irregular supramolecular assembles and cause current spikes or bursts of relatively large conductance. Identification of the structural features that support certain types of membrane currents by various $A\beta$ species enhance our understanding of the molecular basis for membrane permeabilization and cytotoxicity of the most important $A\beta$ peptides involved AD etiology.

## Results

### Membrane channel forming activities of the peptides
Voltage clamp experiments have been conducted at selected hold voltages between +100 mV and −100 mV (the sign corresponds to the trans side, where the peptide was added). At +50 mV transmembrane voltage, the current reached a baseline level of ~40 pA at ~5 min following addition of $A\beta_{1-42}$, featuring stepwise transitions between discrete conductance levels (Fig. 1a). The stability of the conductance pattern and the consistent amplitude of the transitions suggest that $A\beta_{1-42}$ inserts into the membrane and forms ion-conducting channels that are able to switch between on/off states.

A similar behavior of stepwise current transitions was observed for $A\beta_{1-42}$ at other voltages, although the step-like pattern was superimposed with gradual changes of the total conductance level, possibly indicating the presence of a heterogeneous set of ion-conducting structures such as assemblies of different oligomeric numbers (Supplementary Fig. 1), summarized in the histogram (Fig. 1e).

$A\beta_{1-40}$ displayed a different behavior; infrequent burst-like current spikes with <50 ms duration were detected that were superimposed on a zero current level (Fig. 1b). The amplitude, duration, and the frequency of current spikes were significantly higher at 100 mV and -100 mV applied voltages, more frequent at -100 mV than +100 mV, and their magnitude exceeded severalfold the current steps induced by $A\beta_{1-42}$ (Supplementary Fig. 2). Non-zero macro current was not observed for this peptide. The conductance histogram of $A\beta_{1-40}$ featured a peak around the zero level and a relatively wide range for conductance distribution (Fig. 1f).

Membrane conductance induced by $A\beta pE_{3-42}$ involved both stepwise and high frequency burst-like patterns (Fig. 1c, Supplementary Fig. 3). In addition, events of sudden jump between different macro conductance levels occurred, which might result from cooperative opening/closing of multiple channels (Supplementary Fig. 3). Some unique patterns of $A\beta pE_{3-42}$ were the presence of constant macro conductance levels for long dwell time (~1 min) as well as shifts to zero current level, both of which could be overlayed with short bursts of current. This behavior is reflected in the distinct bimodal conductance histogram of this peptide (Fig. 1g, Supplementary Fig. 3).

$A\beta pE_{3-40}$ also exhibited a combination of step-like and burst-like activities, with higher frequency and higher conductance compared to $A\beta_{1-40}$ and $A\beta pE_{3-42}$ (Fig. 1d). The burst-like activities of $A\beta pE_{3-40}$ were continuous throughout the recording and showed similar behavior at all tested voltages. The conductance histogram of $A\beta pE_{3-40}$ was spread over a range of values, with no clearly discernible peaks (Fig. 1h, Supplementary Fig. 4). Even though clear step-like transitions in the conductance were observed for some voltages, rapid switching between sub-states of intermittent conductance also occurred. The change in voltage did not alter the kinetic behavior of the channels, and the frequency of current transition events remained constant.

### Single channel Properties
A close-up view of current traces revealed the single channel properties. $A\beta_{1-42}$ channels switched between closed and open states, designated $L_c$ and $L_o$, respectively, with channel conductance of ~100 pS (Fig. 2a). The open state dwell time varied in a wide range, from < 100 ms to >1 s, indicating slow on/off kinetics. The other peptides displayed more than one open sub-state, designated $L_{o1}$, $L_{o2}$, etc. (Fig. 2). $A\beta_{1-40}$ mostly exhibited burst-like activity, i.e., jumps from $L_c$ to a short-lived $L_{o1}$ state (80-100 pS) and back, although other states of larger conductance (~340 pS) are possible (Fig. 2b). Thus, $A\beta_{1-40}$ does not form stable channels with significant open state dwell time under these conditions.

$A\beta pE_{3-42}$ showed regular switching between discrete states $L_c$ and $L_{o2}$ (~300 pS) while also transitioning to a more seldom intermediate

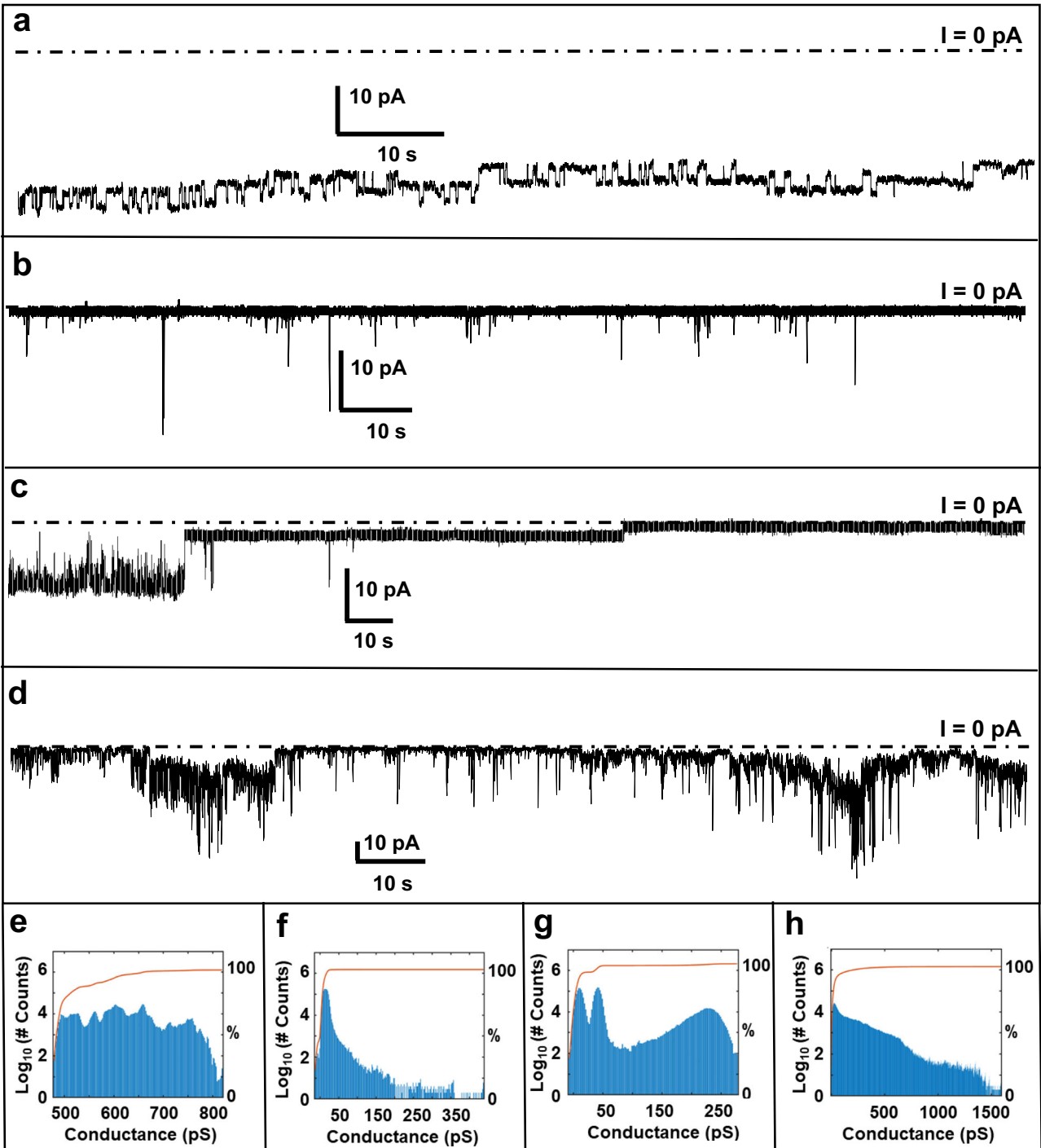

**Fig. 1 | Voltage clamp current traces and conductance histograms.** $A\beta_{1-42}$ (**a**), $A\beta_{1-40}$ (**b**), $A\beta pE_{3-42}$ (**c**), and $A\beta pE_{3-40}$ (**d**) at 50 mV membrane-hold potential in 1 M KCl + 10 mM HEPES buffer, pH 7.4. The membranes were composed of 1-palmitoyl-2-oleoyl-phosphatidylcholine (POPC), 1-palmitoyl-2-oleoyl-phosphatidyglycerol (POPG), and cholesterol (6:3:1 molar ratio). The dotted horizontal line indicates the zero current baseline level. Conductance histograms for $A\beta_{1-42}$, $A\beta_{1-40}$, $A\beta pE_{3-42}$, and $A\beta pE_{3-40}$ are shown in (**e**–**h**) respectively, where the total counts/bin are expressed in log scale in the left y-axis (blue bars) and the cumulative counts (0–100%) per conductance level are indicated in the right y-axis (red line). At least five independent experiments have been conducted with similar results.

states $L_{o1}$ (~150 pS) (Fig. 2c). This peptide showed relatively fast kinetics of transitions between states in the beginning of the trace followed by periods of very long (>1 s) open state dwell time, resembling the behavior of $A\beta_{1-42}$. $A\beta pE_{3-40}$ displayed a complex behavior, i.e., twitching between at least 4 open states ($L_{o1}$ through $L_{o4}$) with conductance values of ~200 pS, 400 pS, 640 pS and 1040 pS, respectively (Fig. 2d). In contrast to the other three peptides, $A\beta pE_{3-40}$ showed continuously transitioning events with relatively fast kinetics.

## Channel Blocking by $Zn^{2+}$ Ions

Since $Zn^{2+}$ ions block Aβ-induced membrane currents (see Introduction), the ability of $Zn^{2+}$ to modulate Aβ channels has been studied. At +100 mV hold voltage, $A\beta_{1-42}$ induced current of ~65 pA, followed by single channel activity (Supplementary Fig. 5a). After first addition of 10 mM $ZnCl_2$, the single channel activity persisted for roughly 40 s, then ceased as the membrane switched to a lower macro conductance level. The current did not fully subside even after a second addition of

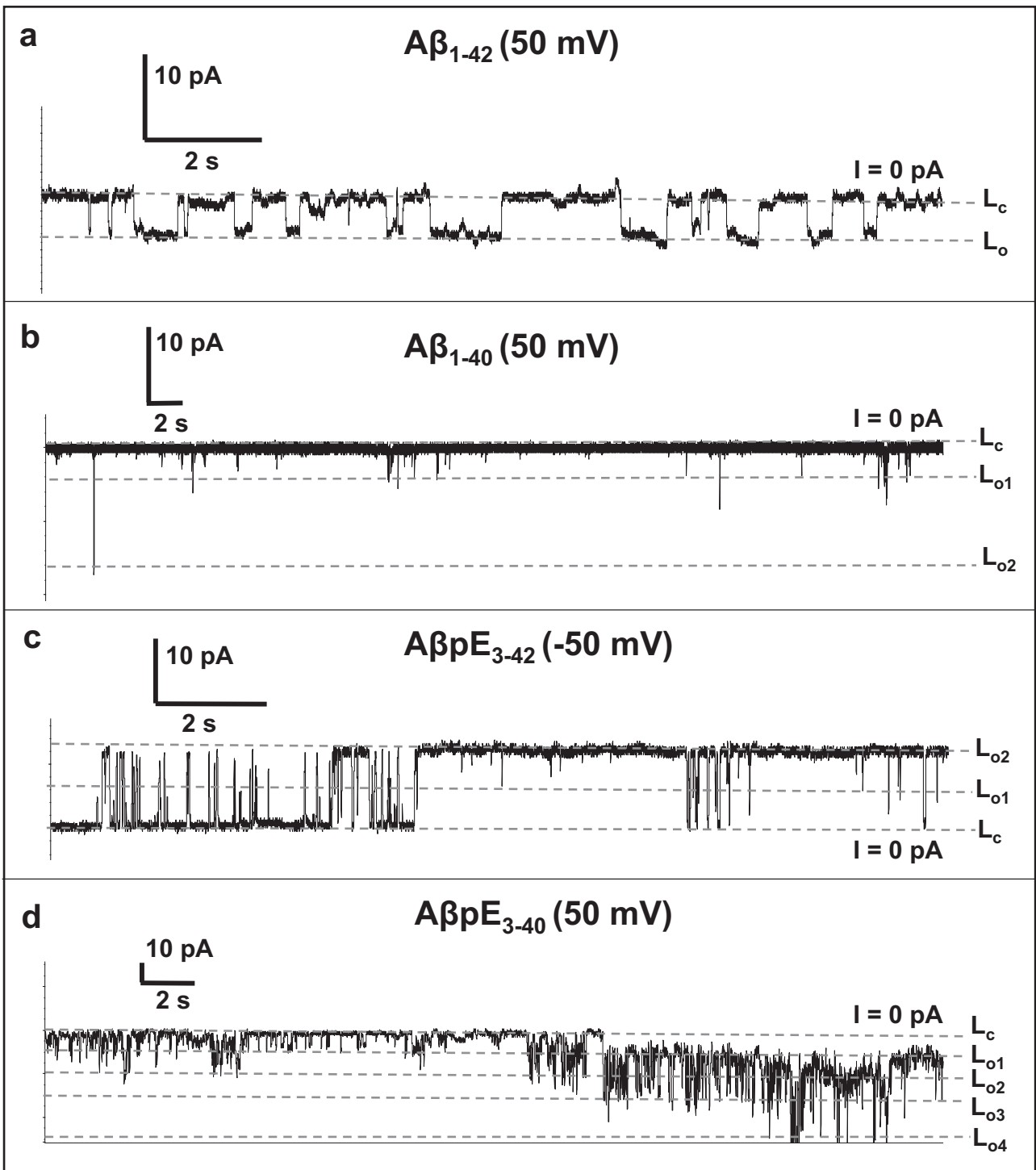

**Fig. 2 | Electrophysiological recordings.** Current traces for Aβ$_{1-42}$ (**a**), Aβ$_{1-40}$ (**b**), AβpE$_{3-42}$ (**c**), and AβpE$_{3-40}$ (**d**) in shorter time scale reveal the single channel conductance behaviors. Discrete conductance levels have been marked with notation L$_c$, i.e. closed state, and L$_{o1}$, L$_{o2}$, ..., corresponding to multiple open state levels. These levels have been assigned to include maximum number of points at a particular conductance level. In all cases except panel **c** (AβpE$_{3-42}$), the hold voltage was +50 mV. For AβpE$_{3-42}$, -50 mV was chosen as the step-like conductance activity was more distinguishable compared to the +50 mV trace. Membrane lipid composition was the same as in Fig. 1, and the buffer was 1 M KCl + 10 mM HEPES, pH 7.4. At least five independent experiments have been conducted with similar results.

10 mM ZnCl$_2$, although most of the flickering activity stopped. Thus, Zn$^{2+}$ does exert an inhibitory effect on Aβ$_{1-42}$ channels but a fraction of ion-conducting structures remains active, consistent with a heterogeneous assembly of Aβ$_{1-42}$ molecules in the membrane.

Aβ$_{1-40}$ showed the characteristic burst-like activity, which ceased ~30 s after addition of 10 mM ZnCl$_2$, followed by an unusual behavior of the current, i.e., appearance of long-lived current flickers (or closely spaced short bursts) in addition to infrequent single bursts. This "anomalous" behavior of the current trace has only been seen for Aβ$_{1-40}$ but neither for the other three peptides nor for membranes without peptide addition. Hence, it reflects the unique structural and membrane interaction properties of Aβ$_{1-40}$, as discussed in the forthcoming sections. At the end of the trace, the current vanished, hence no more ZnCl$_2$ was added (Supplementary Fig. 5b).

AβpE$_{3-42}$ showed large macro conductance, which subsided significantly with the first addition of 10 mM ZnCl$_2$. Following a second addition of ZnCl$_2$, the residual conductance persisted for ~4 min and then stopped, with current returning to the zero baseline level (Supplementary Fig. 5c). AβpE$_{3-40}$ displayed continuous, high frequency burst-like activity (Supplementary Fig. 5d). The first 10 mM ZnCl$_2$ addition reduced the amplitude of the spikes, but higher conductance resumed after ~3 min. Approximately 4 min after second addition of 10 mM ZnCl$_2$, the current died out.

## Peptide Structure from Circular Dichroism

Significant differences in the channel activities of the four Aβ peptides are likely due to their distinct structural features and modes of interactions with lipid membranes. The secondary structure of the peptides without and with lipid vesicles was probed by far-UV circular dichroism (CD). Because peptide samples with lipid vesicles were studied before and after extrusion through 100 nm pore-size filters to obtain unilamellar vesicles of defined size, lipid-free peptide samples were also studied before and after extrusion. Spectra of Aβ$_{1-42}$ displayed a deep minimum at 217-218 nm irrespective of extrusion, indicating β-sheet structure[52] (Fig. 3a). Extruded samples with lipid showed a weaker minimum at 216 nm and a shoulder around 209 nm, suggesting an α/β-type structure. (Unextruded samples with lipid generated spectra with excessive noise due to strong light scattering by vesicles and therefore are not shown.) Aβ$_{1-40}$ generated minima around 217 and 199 nm before extrusion, implying β-sheet and unordered structures (Fig. 3d), and after extrusion the spectrum displayed one minimum at 217 nm assigned to β-sheet. In contrast to Aβ$_{1-42}$, the presence of lipid caused formation of unordered (rather than α-helical) structure in Aβ$_{1-40}$. Spectra of AβpE$_{3-42}$ and AβpE$_{3-40}$ showed weaker and red-shifted nπ* transitions (219-222 nm) before and after extrusion (Fig. 3g, j). With lipid vesicles, spectra of these peptides exhibited an additional feature around 208 nm, indicating α-helix formation.

Thus, Aβ$_{1-42}$ forms well-defined β-sheet structure, followed by the pyroglutamylated peptides, while Aβ$_{1-40}$ tends to be more unordered. In all cases except Aβ$_{1-40}$, the presence of membranes appears to induce partial α-helical structure. Together with the channel data, these results indicate a correlation between β-sheet propensity and step-like channel forming potency.

In the near-UV region, the CD spectra displayed a negative band at 278-282 nm, which is the induced CD signal due to the S$_0$→S$_1$ (ground to first excited singlet state) transition in the aromatic amino acid

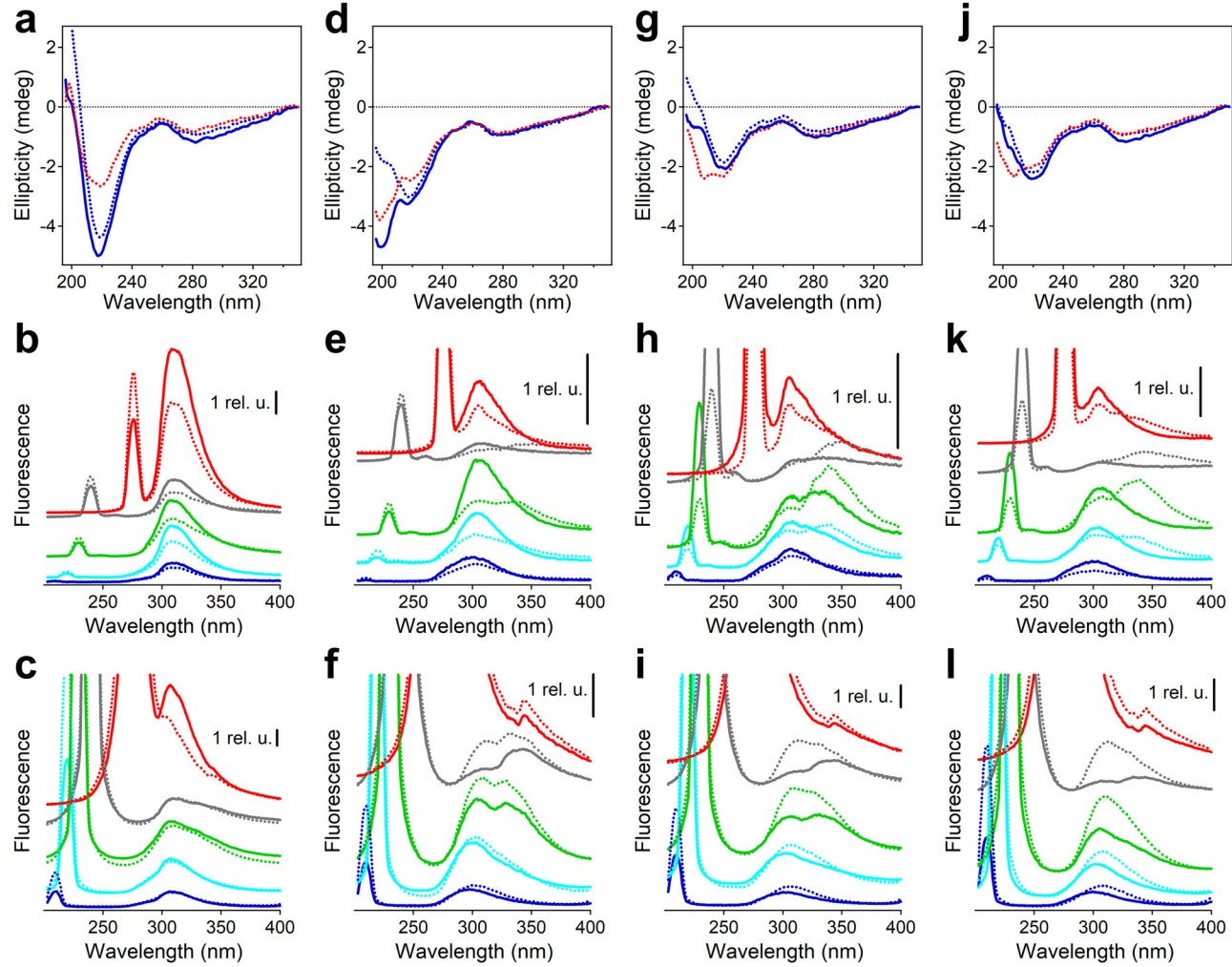

**Fig. 3 | CD and fluorescence spectra.** Aβ$_{1-42}$ (**a–c**), Aβ$_{1-40}$ (**d–f**), AβpE$_{3-42}$ (**g–i**), and AβpE$_{3-40}$ (**j–l**) with or without lipid. In CD spectra (upper row), blue and red lines correspond to the absence and presence of lipid vesicles, respectively (POPC:POPG:cholesterol at 6:3:1 molar ratio). For fluorescence data, the middle row presents spectra in the absence of lipid and the lower row shows data in the presence of lipid vesicles. The excitation was at 210 nm (blue), 220 nm (cyan), 230 nm (green), 240 nm (gray), and 275 nm (red). Dotted lines indicate that the samples have been extruded through 100 nm pore-size polycarbonate membranes for both CD and fluorescence spectra. Peptide and total lipid (when present) concentrations were 20 μM and 1 mM, respectively, the buffer was 25 mM NaCl + 25 mM Na,K-phosphate, pH 7.2, and the temperature was 20 °C. Three independent experiments have been conducted with similar results.

side chains[53]. This CD band did not depend on the presence of lipid. However, fluorescence spectroscopy revealed interesting structural features of the peptides, described in the next section.

## Structural features from fluorescence spectroscopy

As described in Supplementary Fig. 6, excitation of $A\beta_{1-42}$ at wavelength ($\lambda_{ex}$) varying from 200 nm to 380 nm results in two effects, Rayleigh scattering at $\lambda_{ex}$ and fluorescence at two regions of $\lambda_{ex}$, i.e., 220-230 nm and 260-280 nm, due to $S_0 \rightarrow S_2$ and $S_0 \rightarrow S_1$ transitions, respectively. These transitions are also known as $^1L_a$ and $^1L_b$, respectively, according to Platt designation[54] and are schematically illustrated in Supplementary Fig. 7. $A\beta_{1-42}$ produced strong tyrosine (Tyr) fluorescence at 307-310 nm with $\lambda_{ex}$ varying between 210 nm and 275 nm before and after extrusion (Fig. 3b). Similar emission wavelengths following $S_0 \rightarrow S_2$ and $S_0 \rightarrow S_1$ excitations indicate $S_2 \rightarrow S_1$ internal conversion and emission from a low vibrational energy state of $S_1$ in both cases, consistent with Kasha's rule (see Supplementary Fig. 7). $A\beta_{1-40}$ displayed weaker and blue-shifted fluorescence (301-307 nm), and the blue shift was stronger at shorter $\lambda_{ex}$ (Fig. 3e). These features suggest that phenylalanine (Phe) contributes to fluorescence more than in case of $A\beta_{1-42}$ because Tyr of $A\beta_{1-40}$ is exposed to and quenched by the buffer[55]. For the extruded samples of $A\beta_{1-40}$, an interesting effect has been detected. At $\lambda_{ex} = 230$ nm, the emission band was split into peaks around 306 and 340 nm, and with $\lambda_{ex} = 240$ nm the new, red-shifted component extended to larger wavelengths. The red-shifted component of Tyr emission can be rationalized in terms of a specific solvent effect, i.e., strong H-bonding of the phenolic OH group to a base such as $HPO_4^{2-}$ and, possibly, deprotonation. Extrusion rendered a large fraction of the Tyr of $A\beta_{1-40}$ exposed to the buffer, resulting in red-shifted emission. The lack of such splitting in case of $A\beta_{1-42}$ indicates this peptide forms a more compact, solvent-protected β-sheet structure before and after extrusion.

Spectra of $A\beta pE_{3-42}$ were split, with the red-shifted component around 340 nm (for $\lambda_{ex} = 220$ nm and 230 nm) or 350 nm (for $\lambda_{ex} = 240$ nm) in addition to the lower-wavelength component at 306-310 nm, and the effect was more pronounced for the extruded samples (Fig. 3h). $A\beta pE_{3-40}$ generated one emission band at 300-308 nm before extrusion and split bands after extrusion, with a red-shifter component at 333-342 nm (Fig. 3k). The absence of the red-shifted emission at $\lambda_{ex} = 275$ nm indicates that the $S_0 \rightarrow S_1$ ($^1L_b$) transition is less solvent-sensitive than the $S_0 \rightarrow S_2$ ($^1L_a$) transition because their dipoles are oriented across and along the phenolic ring, respectively, the latter involving the polar hydroxyl group[55], as shown in Supplementary Fig. 7. Thus, the whole process where deprotonation is involved is thought to proceed as follows: $S_0 \rightarrow S_2$ transition, excited state deprotonation facilitated by a proton acceptor, $S_2 \rightarrow S_1$ internal conversion, vibrational decay to the lowest energy level of $S_1$, emission, i.e. radiative transition to a vibrational level of $S_0$. These data indicate that (a) splitting of Tyr fluorescence can be used to assess solvent exposure, and hence the compactness of the tertiary fold, of proteins, (b) fluorescence of solvent-exposed and protected fractions can be excited selectively, and (c) the $Tyr_{10}$ residue is most solvent-exposed in $A\beta pE_{3-42}$ and most solvent-protected in $A\beta_{1-42}$, the other two peptides showing intermediate solvent exposure of $Tyr_{10}$.

The presence of lipid vesicles resulted in much stronger Rayleigh scattering (Fig. 3c, f, i, l). Fluorescence spectra of $A\beta_{1-42}$ did not undergo splitting (Fig. 3c), as in the absence of lipid, indicating a compact tertiary fold. In case of $A\beta_{1-40}$, the emission was split at $\lambda_{ex} = 230$ nm and 240 nm both before and after extrusion (a red-shifted component at 330-344 nm in addition to one at 304-313 nm) (Fig. 3f). This means a solvent-exposed Tyr of $A\beta_{1-40}$ even in the presence of vesicles. A poor membrane insertion ability of $A\beta_{1-40}$ is consistent with its less efficient channel forming activity, as discussed above. A solvent-accessible structure in the presence of lipid before and after extrusion was also exhibited by $A\beta pE_{3-42}$, although in this case the red-

shifted component was less intense and was absent at $\lambda_{ex} = 220$ nm (Fig. 3i), suggesting partial membrane insertion. Spectra of $A\beta pE_{3-40}$ in the presence of lipid did not show splitting, especially for the extruded samples (Fig. 3l), implying a more efficient protection from the solvent by the membranes. The fluorescence of $A\beta_{1-40}$, $A\beta pE_{3-42}$, and $A\beta pE_{3-40}$ with excitation at 275 nm was obscured by strong Rayleigh scattering; the feature around 344 nm is most likely the resonance-Raman scattering peak (Fig. 3f, i, l). This underscores the usefulness of using a lower wavelength excitation as it not only clears the spectral window of Tyr emission but also reveals the degree of solvent exposure of the fluorophore.

Tyr emission splitting may be facilitated by the solvent but also by amino acid side chains with proton acceptor properties such as aspartate or glutamate[56]. In the latter case, the effect would be unaltered upon change of the buffer. To test this conjecture, experiments have been conducted in a Tris-HCl buffer. Data of Supplementary Fig. 8 show that the Tyr emission splitting is not unique to the phosphate, but the effect is stronger in phosphate than in Tris buffer (see more detail in the legend of Supplementary Fig. 8). Conceivably, Tris can weaken the OH bond of Tyr side chain by H-bonding with both hydrogen and oxygen atoms by its hydroxyl and amino groups but to a lesser extent than $HPO_4^{2-}$. CD spectra of all four peptides in Tris buffer were similar to those in phosphate buffer (see Fig. 3 upper row and Supplementary Fig. 9), indicating no secondary structural differences, as expected. Splitting of Tyr emission also occurred in an unbuffered solution of 25 mM NaCl, although with a smaller red-shift effect (Supplementary Fig. 10). This is exemplified for $A\beta pE_{3-42}$ at $\lambda_{ex} = 230$ nm: the red-shifted component was located at 338-340 nm in phosphate and Tris buffers and at 327-336 nm in the unbuffered solution (cf. green lines in Fig. 3h, Supplementary Figs. 8e and 10). Also, the intensity of the red-shifted component for the extruded peptide was much stronger in phosphate buffer than in Tris buffer and in the unbuffered solution. While the involvement of amino acids with proton acceptor groups cannot be ruled out, clear differences between phosphate and Tris buffers and the unbuffered solution suggest that the splitting of Tyr fluorescence is caused by the solvent and is stronger when strong H-bonding acceptors and donors are present in the buffer.

The structural features of the peptides have been assessed in additional experiments at 37 °C to ensure those features are maintained at a physiological temperature. Both CD and fluorescence data showed that the peptides' secondary structure and Tyr fluorescence splitting features at 37 °C were similar to those seen at 20 °C. Details are presented in the legends to Supplementary Figs. 11 and 12.

Overall, fluorescence and CD data indicate that $A\beta_{1-42}$ forms tightly packed β-sheet structure and in membranes acquires a fraction of α-helix. $A\beta_{1-40}$ forms less stable and more solvent-exposed β-sheet structure, and the vesicles induce partially unordered structure without protecting from solvent, indicating its poor ability to insert into the membranes. $A\beta pE_{3-42}$ and $A\beta pE_{3-40}$ form β-sheet structure in buffer and solvent-protected α/β structure in membranes, indicating membrane insertion. Thus, a correlation is established between the β-sheet propensity, membrane insertion, and channel forming capabilities of the four Aβ peptides.

## Structure and orientation of the peptides in membranes from polarized ATR-FTIR spectroscopy

Polarized attenuate total reflection Fourier transform infrared (ATR-FTIR) spectroscopy was used to gain information on the secondary structure of the peptides reconstituted in lipid multilayers and the orientation relative to the membrane. Figure 4 shows the ATR-FTIR spectra of all four peptides at parallel and perpendicular polarizations of the incident light with respect to the plane of incidence along with the spectral components, as well as the dichroic spectra, in the lipid carbonyl stretching and the peptides' amide I regions. The lipid

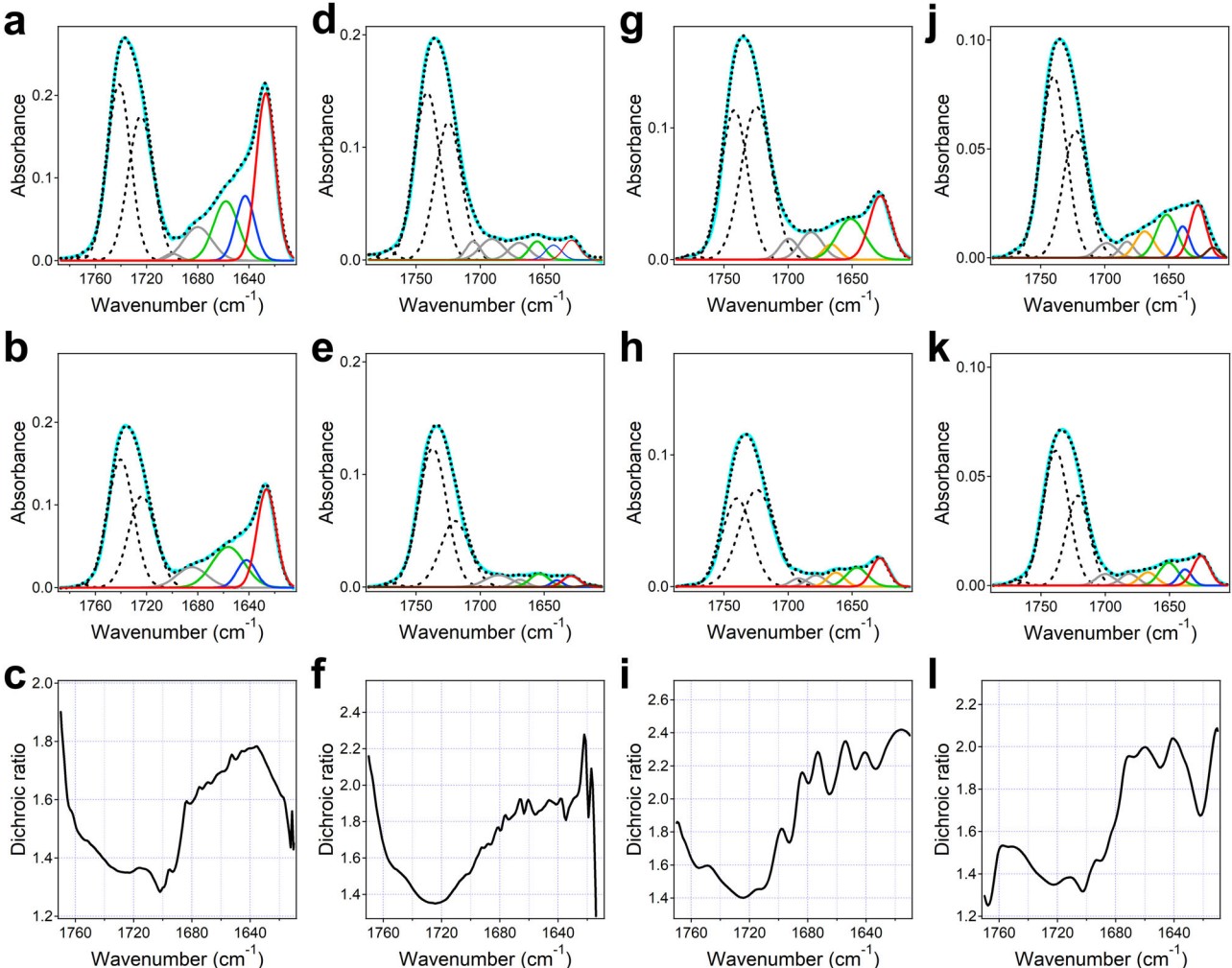

**Fig. 4 | ATR-FTIR spectra of the peptides.** $A\beta_{1-42}$ (**a, b**), $A\beta_{1-40}$ (**d, e**), $A\beta pE_{3-42}$ (**g, h**) and $A\beta pE_{3-40}$ (**j, k**) reconstituted in lipid multilayers composed of 60 mol % POPC, 30 mol % POPG, and 10 mol % cholesterol at 1:50 peptide-to-total lipid molar ratio, hydrated by a $D_2O$-based buffer (25 mM NaCl, 25 mM Na,K-phosphate, pD 7.2), at parallel (first row) and perpendicular (second row) polarizations of the incident light relative to the plane of incidence. The measured spectrum is shown in solid line colored cyan, and the fitted curve is shown as black dotted line. Spectral components in lipid carbonyl and peptide amide I regions are presented as follows: lipid carbonyl components: black dashed lines; turn structures: gray; $\alpha_{II}$-helix: orange; $\alpha$-helix: green; unordered: blue; $\beta$-sheet: red; side chains: brown. Panels (**c–f–i–l**) show the dichroic spectra, i.e., the ratio of spectra at II and ⊥ polarizations, for $A\beta_{1-42}$, $A\beta_{1-40}$, $A\beta pE_{3-42}$, and $A\beta pE_{3-40}$, respectively. Three independent experiments have been conducted with similar results.

carbonyl group stretching vibration generates an absorbance band with two components at 1742 $cm^{-1}$ and 1728 $cm^{-1}$, which correspond to dehydrated and hydrated C = O groups, respectively; H-bonding with water weakens the C = O covalent bond and thereby decreases the vibrational frequency[57].

Significant differences have been detected between the secondary structures of the peptides in supported membranes. $A\beta_{1-42}$ displayed ~46% β-sheet and ~19% α-helix structures, the rest of the peptide being in turn, irregular, or other conformations (Table 1). In contrast, $A\beta_{1-40}$ mostly adopted turn, irregular, or other conformations (68%) with only ~17% β-sheet and ~15% α-helix. The pyroglutamylated peptides had 27–35% β-sheet and 21–25% α-helix. In reality, a fraction of turns is located between β-strands and should be counted as part of the β-sheet structure. The pyroglutamylated peptides displayed an additional amide I component in the 1666-1663 $cm^{-1}$ region, which has been assigned to $\alpha_{II}$-helix, i.e. a helical structure with tilted amide plane and with weaker helical H-bonding[58]. This $\alpha_{II}$-helical structure constituted around 9–12% (3.5–4.6 amino acid residues), i.e. about one helical turn likely located at the edge of the regular α-helix. Given the spectral overlap of type II β-turn and $\alpha_{II}$-helical structures[59], these ~4

amino acid residues may alternatively be assigned to β-turn structure. These results are consistent with CD data (Fig. 3a, d, g, j and Supplementary Figs. 9, 11), indicating an ordered, mostly β-sheet structure of $A\beta_{1-42}$, mostly unordered (or turn) conformation of $A\beta_{1-40}$, and α/β structure of $A\beta pE_{3-42}$ and $A\beta pE_{3-42}$ in lipid membranes.

Notably, the amide I intensity of $A\beta_{1-40}$ relative to that of the lipid C = O band is much weaker compared to $A\beta_{1-42}$ (Fig. 4a, b vs. d, e), indicating that a significant part of $A\beta_{1-40}$ is washed away during injection of the buffer owing to its more hydrophilic nature. $A\beta pE_{3-42}$ and $A\beta pE_{3-40}$ have intermediate amide I intensities. This is consistent with fluorescence data showing a most efficient membrane insertion and solvent protection for $A\beta_{1-42}$ and solvent exposure for $A\beta_{1-40}$ and the pyroglutamylated peptides (Fig. 3, Supplementary Figs. 8, 12).

The average orientation of β-strands of membrane-embedded peptides with respect to the membrane normal was deduced from polarized ATR-FTIR studies and varied between ⟨β⟩ = 30 and 40 degrees (Supplementary Table 1), similar to the respective angle of mitochondrial and bacterial porins, which form β-barrel structure[60]. The α-helices were tilted obliquely at 50-65 degrees relative to the membrane normal.

**Table 1 | Amide I wavenumbers (v), fractions (f), and numbers of amino acid residues (N) for α-helix, β-sheet, unordered (ρ) structures, and turn and "other" structures (other), derived from ATR-FTIR spectra**

| | Aβ$_{1-42}$ | Aβ$_{1-40}$ | AβpE$_{3-42}$ | AβpE$_{3-40}$ |
|---|---|---|---|---|
| $v_\alpha$(cm$^{-1}$) | 1659–1657[a] | 1656–1654 | 1651–1646 / 1666–1663[b] | 1652-1651 / 1668-1666[b] |
| $f_\alpha$ | 0.194 ± 0.0236 | 0.153 ± 0.0170 | 0.254 ± 0.0548 / 0.088 ± 0.0401[b] | 0.212 ± 0.0396 / 0.121 ± 0.0421[b] |
| $N_\alpha$ | 8.15 ± 0.992 | 6.12 ± 0.680 | 10.16 ± 2.193 / 3.52 ± 1.606[b] | 8.06 ± 1.504 / 4.60 ± 1.600[b] |
| $v_\beta$(cm$^{-1}$) | 1627 | 1629 | 1628 | 1627–1625 |
| $f_\beta$ | 0.459 ± 0.0367 | 0.170 ± 0.0190 | 0.351 ± 0.0521 | 0.270 ± 0.0419 |
| $N_\beta$ | 19.27 ± 1.541 | 6.80 ± 0.760 | 14.04 ± 2.0846 | 10.27 ± 1.591 |
| $v_\rho$(cm$^{-1}$) | 1643–1642 | 1642–1640 | 1641–1637 | 1639–1637 |
| $f_\rho$ | 0.188 ± 0.0150 | 0.140 ± 0.0525 | 0.065 ± 0.0596 | 0.176 ± 0.0809 |
| $N_\rho$ | 7.90 ± 0.626 | 5.60 ± 2.100 | 2.60 ± 2.383 | 6.70 ± 3.0747 |
| $v_{other}$(cm$^{-1}$) | 1700–1678 | 1705-1669 | 1699-1678 | 1700-1681 |
| $f_{other}$ | 0.159 ± 0.0151 | 0.537 ± 0.0551 | 0.242 ± 0.0549 | 0.221 ± 0.0405 |
| $N_{other}$ | 6.68 ± 0.632 | 21.48 ± 2.204 | 9.68 ± 2.197 | 8.37 ± 1.539 |

The range of wavenumbers and mean ± standard deviation values have been determined from three independent experiments.

[a]Wavenumbers of ⊥ spectra are usually slightly lower than those of ∥ spectra.

[b]These components have been assigned to α$_{II}$-helix, although β-turn structure is also possible.

## Effect of Aβ peptides on lipid membranes

The lipid acyl chain order parameter was determined from polarized ATR-FTIR spectra in the methylene (CH$_2$) stretching region to (a) ensure a meaningful structural order of membrane lipids and (b) assess the possible effects of the peptides on membrane structure. The CH$_2$ groups undergo asymmetric and symmetric stretching vibrations that generate absorbance bands around 2920 cm$^{-1}$ and 2850 cm$^{-1}$, respectively[61] (Supplementary Fig. 13). As seen from Supplementary Table 2, the order parameter of plain lipid was ~0.44, indicating well organized membranes (see legend to Supplementary Fig. 13 for justification). The presence of peptides in the membranes exerted non-trivial effects on the lipid order. Aβ$_{1-42}$ increased the lipid order whereas the other three peptides decreased $S_L$ (Supplementary Table 2). The difference between the effect of Aβ$_{1-42}$ and the other three peptides was statistically significant as judged from the p-values when comparing $S_L$ in the presence of Aβ$_{1-42}$ with those in the presence of Aβ$_{1-40}$, AβpE$_{3-42}$, and AβpE$_{3-40}$ ($p = 0.0468, 0.0374$, and $0.0089$, respectively). This finding suggests a unique property of Aβ$_{1-42}$ compared to the other peptides, which may be related to its ability to form stable ion-conducting channels in membranes. Aβ$_{1-40}$ and AβpE$_{3-40}$, on the other hand, decreased the lipid order, an effect possibly related to a membrane destabilization mechanism of ion conductance induced by these peptides.

## Morphology of Aβ peptides in lipid membranes from AFM

The morphological features of the four Aβ peptides without and with reconstitution in lipid membranes were probed by AFM. First, peptides freshly suspended in aqueous buffer and incubated under fibrillization conditions (24 h, 37 °C) were studied. All freshly suspended peptides showed a distribution of monomers and oligomers in the height range of 1–5 nm (Supplementary Fig. 14). Aβ$_{1-42}$ and Aβ$_{1-40}$ showed a uniform distribution of monomers and oligomers in the height range 1–3 nm (section profile below Supplementary Fig. 14a, b) while AβpE$_{3-42}$ and AβpE$_{3-40}$ displayed higher ordered oligomeric structures with heights ranging from 2 nm to 5 nm above the mica plane (section profile below Supplementary Fig. 14c, d).

Under fibrillogenesis conditions, Aβ$_{1-42}$ and Aβ$_{1-40}$ showed highly dense, entangled fibrous structures (Supplementary Fig. 14e, f). In contrast, the pyroglutamylated peptides showed a very sparse distribution of individual fibrils without entanglement (Supplementary Fig. 14g, h). Even though all peptides showed a positive thioflavin-T signal, the fibril distribution for the pyroglutamylated variants was not easily visualized as compared to the unmodified peptides and some higher order, large-sized oligomers could be observed, which may indicate incomplete fibril elongation.

Next, the morphological features of the peptides reconstituted in lipid membranes were examined (Fig. 5). In all cases, the image analysis showed two populations of peptide oligomers, one protruding from the membrane 0.5–2.0 nm above the bilayer surface, assigned to membrane-inserted structures, and the other exceeding 2 nm, assigned to a membrane-adsorbed peptide pool (Fig. 5e–h).

Among the membrane inserted fractions, a small sub-population exhibited pore-like morphology. These structures were better resolved in case of Aβ$_{1-42}$ and featured annular assemblies composed of 4–6 units with an outer diameter of ~16 nm (Fig. 5i). The other three peptides exhibited oligomeric structures in comparable height ranges, although the histogram of Aβ$_{1-40}$ was shifted towards larger heights above membrane surface (Fig. 5f), consistent with membrane-adsorbed rather that embedded mode of membrane binding of this peptide. Pore-like morphologies were not identified for Aβ$_{1-40}$ and AβpE$_{3-40}$ (Fig. 5j, l). In the case of AβpE$_{3-40}$, although an apparent subunit like topology was weakly observed (Fig. 5l), AFM tip convolution effects prevented from fully resolving the subunits. For AβpE$_{3-42}$, annular, pore-like structures were seen with outer diameter from 9 to 15 nm (Fig. 5k). Clusters of pore-like structures were seen for this peptide as well. Overall, apparent pore-like structures were resolved more readily for Aβ$_{1-42}$, followed by AβpE$_{3-42}$, consistent with the patch clamp data showing regular step-like channel behavior for these two peptides.

## Discussion

In this work, membrane channel formation by the most abundant and toxic forms of Aβ peptides, i.e., Aβ$_{1-42}$, Aβ$_{1-40}$, AβpE$_{3-42}$, and AβE$_{3-40}$, have been analyzed and correlated with their structural and morphological features. The peptides have been reconstituted in lipid membranes composed of POPC, POPG, and cholesterol to mimic the fluidity, the anionic surface charge, and cholesterol content of neuronal membranes[62]. In fact, channel conductances recorded in this work range from 100 pS to 1000 pS, similar to those induced by Aβ peptides in cell membranes[25,30]. Analysis of the peptides under identical conditions allowed identification of a clear correlation between the β-sheet content, degree of membrane insertion, morphological features, and channel formation abilities of the peptides.

Aβ$_{1-42}$ and Aβ$_{1-40}$, which differ by only two amino acids (Ile$_{41}$Ala$_{42}$), exhibit strikingly diverse structural and channel forming properties. Aβ$_{1-42}$ contains the largest β-sheet fraction (2.7-fold more than Aβ$_{1-40}$), efficiently inserts into lipid membranes, forms annular supramolecular assemblies, and produces single-channel-like currents resembling those observed in cell membranes[22,30]. Aβ$_{1-40}$, on the other hand, displays a conformation rich in turn and unordered structures, with little β-sheet, fails to effectively embed in membranes, shows irregular morphology, and produces infrequent bursts of current, again similar to those detected in neuronal membranes[25]. These data lead to a conclusion that Ile$_{41}$ and Ala$_{42}$ play a crucial role in promoting β-sheet formation and membrane insertion of the Aβ peptides and subsequent channel formation. This is consistent with effective ion channel formation in membranes of HEK293 cells by Aβ$_{1-42}$ but not Aβ$_{1-40}$, which was attributed to differences in membrane interaction and insertion abilities of the two Aβ species[30]. Given the conflicting data reported for membrane pore formation by Aβ$_{1-40}$ (see Introduction), our results argue in favor of perturbation of cellular or lipid membranes by this peptide via a mechanism different from regular channels that switch between open and closed states. This work identifies another

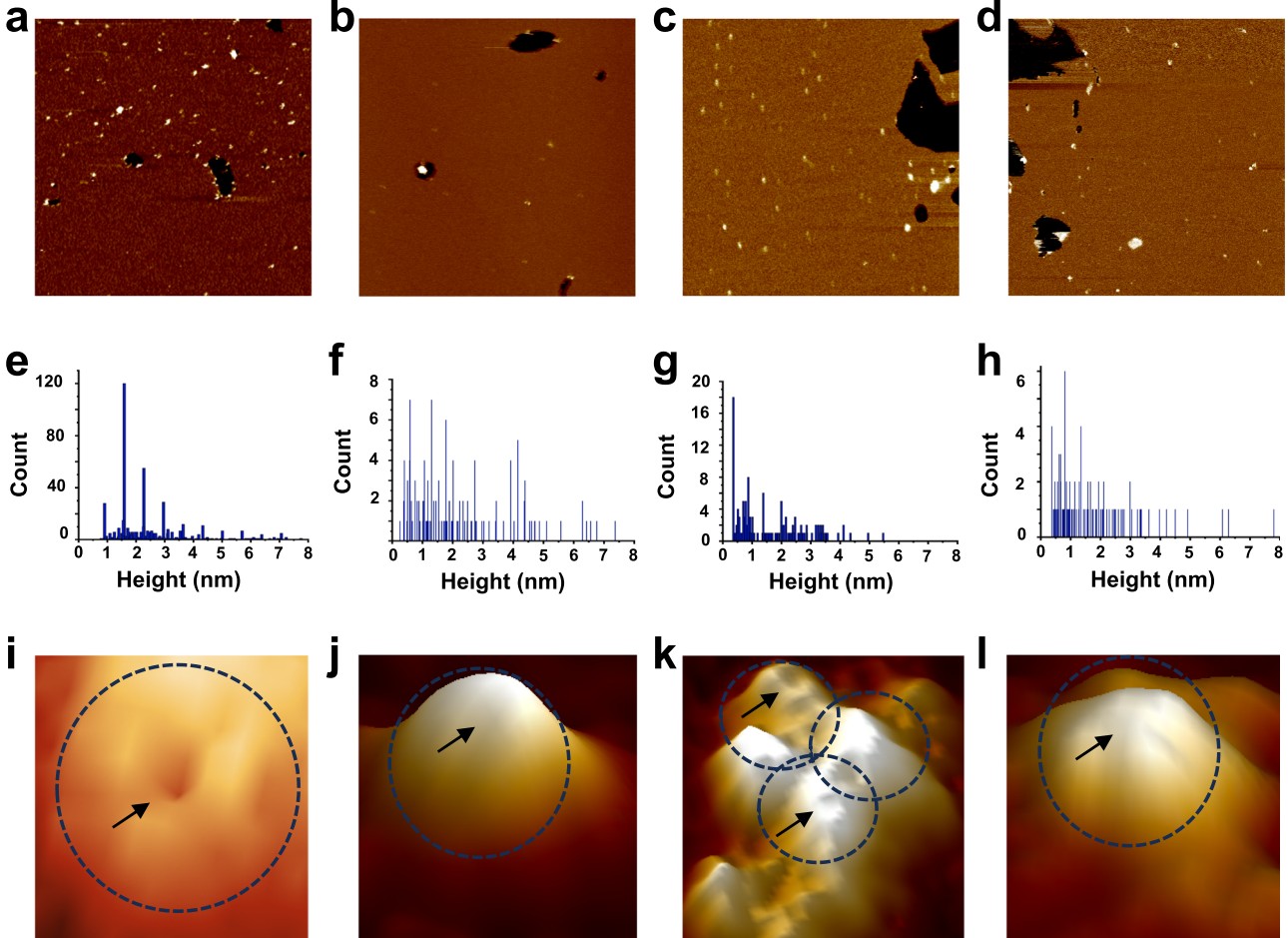

**Fig. 5 | AFM height images of the peptides reconstituted in POPC:POPG:cholesterol (6:3:1 molar ratio) membranes at 1:500 peptide:lipid molar ratio.** (**a**) Aβ$_{1-42}$ (scan size = 1.6 μm × 1.6 μm), (**b**) Aβ$_{1-40}$ (scan size = 1.5 μm × 1.5 μm), (**c**) AβpE$_{3-42}$ (scan size = 2.0 μm × 2.0 μm), (**d**) AβpE$_{3-40}$ (scan size = 1.7 μm × 1.7 μm). For the upper-row images, the darkest part of the image corresponds to the mica surface, and the z-scale is 0–5 nm. The center row presents histograms of particle height above the bilayer plane for Aβ$_{1-42}$ (**e**) Aβ$_{1-40}$ (**f**) AβpE$_{3-42}$ (**g**) AβpE$_{3-40}$ (**h**) combined from 5 separate images per peptide. The lower row presents close-up 3D views of ion channel/pore-like structures formed by Aβ$_{1-42}$ (panel (**i**) scan size = 20 nm × 20 nm), Aβ$_{1-40}$ (panel (**j**) scan size = 30 nm × 30 nm), AβpE$_{3-42}$ (panel (**k**) scan size = 30 nm × 30 nm), and AβpE$_{3-40}$ (panel (**l**) scan sizes = 20 nm × 20 nm). For all high-resolution images (lower row), the z-scale is 0–2 nm. The dashed circles represent structures resembling ion channels and the arrows indicate the position of the pore. The buffer was 300 mM KCl and 10 mM HEPES, pH 7.4. Three independent experiments have been conducted with similar results.

distinction between Aβ$_{1-42}$ and Aβ$_{1-40}$: Aβ$_{1-42}$ increases whereas Aβ$_{1-40}$ decreases the structural order of lipid acyl chains (Supplementary Table 2). Conceivably, the regular β-barrel-like channel structures formed by Aβ$_{1-42}$ stabilizes the structure of surrounding lipids while Aβ$_{1-40}$ exerts an opposite, membrane destabilizing effect, which possibly contributes to induction of irregular current spikes.

The pyroglutamylated peptides show a combination of stepwise and burst-like current patterns. Notably, these peptides are able to form pores of larger conductance compared to their unmodified counterparts. An earlier study showed that 74% and 21% of AβpE$_{3-42}$ formed channels in lipid bilayers with conductance < 100 pS and between 100 pS and 200 pS, respectively, while for Aβ$_{1-42}$ these percentages were 91% and 5%, indicating overall larger channels formed by AβpE$_{3-42}$[39]. Consistent with this, the conductance of L$_{o2}$ state of AβpE$_{3-42}$, observed in this work, is 300 pS, much larger than the 100 pS channels formed by Aβ$_{1-42}$. Channel data for AβpE$_{3-40}$ are not available, but our data indicate this peptide is able to form even larger channels, with conductance reaching 1 nS. It is interesting to note in this context that AβpE$_{3-40}$ exerted stronger cytotoxic effect on cultured rat hippocampal neurons and cortical astrocytes than Aβ$_{1-40}$, Aβ$_{1-42}$, or AβpE$_{3-42}$[63], which might be related to its ability to form larger channels in cell membranes. The high frequency of opening of relatively large channels formed by AβpE$_{3-42}$

and AβpE$_{3-40}$ is likely to contribute to their augmented cytotoxicity, partially confirmed by induction of more efficient Ca$^{2+}$ influx into neurons caused by AβpE$_{3-42}$ as compared to Aβ$_{1-42}$[7].

Channel internal diameters ($d$) were estimated using the measured single channel conductance values and the structural data, as described in the Supplementary Methods and presented in the Supplementary Table 3. For each peptide, a specific channel length ($l$) has been determined based on data of Table 1 and Supplementary Table 1. As the best single channel forming peptides, such as Aβ$_{1-42}$, contained large fractions of β-sheet, with the β-strands tilted from the membrane normal at an angle of 30-40 degrees, these simulations have been conducted assuming a β-barrel-like structure for the peptides. Two values for the resistivity of the salt solution were used, the bulk value and the 5-fold elevated "corrected" value, which we consider more realistic as justified in the Supplementary Methods. The channel diameters, using the corrected solution resistivities, were around 0.56 nm for Aβ$_{1-42}$ channels with 100 pS conductance and around 0.57 nm and 0.83 nm for AβpE$_{3-42}$ channels with 150 pS and 300 pS conductance (Supplementary Table 3). For AβpE$_{3-40}$ channel conductances of 200 pS, 400 pS, 640 pS, and 1040 pS, we obtained $d$ = 0.59 nm, 0.86 nm, 1.12 nm, and 1.48 nm, respectively. Using the bulk resistivity value resulted in nearly 2-fold narrower pore diameters. Aβ$_{1-40}$ showed

a minimal β-sheet fraction, corresponding to ~7 amino acid residues (Table 1), which, when coupled with the strand tilt angle of ~35°, can make a 2 nm long barrel-like structure, about half of the membrane thickness. Based on these results, $A\beta_{1-42}$ is likely to form stable transmembrane channels, consistent with clear step-like current levels and increased lipid order (Supplementary Table 2). $A\beta pE_{3-42}$ may form transmembrane channels as well ($l = 3.84$ nm) with minimal effect on the lipid order, whereas $A\beta pE_{3-40}$ ($l \approx 3$ nm) is likely to be incompletely membrane inserted, exerting a lipid destabilizing effect (Supplementary Table 2). $A\beta_{1-40}$ ($l = 2$ nm) is not likely to form a transmembrane channel, consistent with the unique spiky current features generated by this peptide (Figs. 1, 2). Instead, it may cause membrane permeabilization by the "carpet" or detergent-like mechanisms. Still, $A\beta_{1-40}$ is included in Supplementary Table 3 to show that if it were to form a β-barrel, then its pore diameter would vary between 0.34 and 0.66 nm, using the corrected solution resistivity.

Internal channel diameters for $A\beta_{1-42}$ in lipid bilayers have been evaluated based on electrophysiological recordings, using channel lengths from 3 nm[37] to 6 nm[30,40], and varied between 0.7 nm and 2.1 nm[37,40] whereas in cell membranes they varied between 1.7 nm and 2.4 nm assuming $l = 7$ nm[30]. MD simulations of $A\beta_{1-42}$ and $A\beta pE_{3-42}$ octadecamers yielded pore diameters of up to 2.2 nm (see references in ref. [40]). Channel diameters shown in the Supplementary Table 3 are close to those reported for lipid bilayers but are smaller when compared to $A\beta_{1-42}$ channels in cell plasma membranes[30]. This raises the possibility that cellular components, such as gangliosides or accessory proteins such as the cellular prion protein[32], may facilitate formation of larger Aβ pores.

Interestingly, the $Tyr_{10}$ of $A\beta_{1-42}$ was solvent-inaccessible in the absence and presence of lipid membranes, suggesting that compact secondary and tertiary structures form in the aqueous phase and then the peptide inserts into the membrane and forms regular ion channels. The other three peptides had solvent-exposed $Tyr_{10}$; lipid membranes either failed to protect it from the solvent ($A\beta_{1-40}$) or shielded it partially ($A\beta pE_{3-42}$) or totally ($A\beta pE_{3-40}$), indicating distinct modes of membrane binding/insertion of various Aβ species. Among the four peptides studied here, only $A\beta_{1-42}$, which is the full-length peptide in terms of possessing both the $Asp_1Ala_2$ and $Ile_{41}Ala_{42}$ residues, forms large fraction of β-sheet, effectively inserts into the membrane, displays well-defined pore like morphology, and forms regular ion channels. This is reminiscent of the inability of a N- and C-terminally truncated peptide, $A\beta_{4-34}$, to form pore-like structures and stable pores in lipid membranes[64]. The other three peptides that lack two amino acids at one or both termini have smaller β-sheet content, are more solvent exposed, and induce irregular current bursts, albeit of large conductivity. Thus, our data uncover a delicate relationship between all four structural levels of Aβ peptides and their membrane pore formation capabilities and hence neurotoxic potentials. The presence of the N- and C-terminal residues (primary structure), the β-sheet content (secondary structure), the compactness of the tertiary fold including the N-terminal segment harboring $Tyr_{10}$, and the supramolecular quaternary structure all are inter-related and contribute to the mode of membrane binding and pore formation. While more work needs to be done to provide more detailed insight into the molecular mechanisms of membrane pore formation by these peptides, including the specific roles of the key amino acids and high-resolution structures of Aβ membrane pores by methods such as NMR or cryo-electron microscopy or tomography[32,65], the present data shed light on the intricate relationships between the structural propensities and membrane permeabilization features by these most important Aβ peptides.

## Methods
Information about materials used and the full protocols are available in the Supplementary Methods. Here, a brief description of all procedures is presented.

## Voltage clamp electrophysiology experiments
The lipid bilayers were prepared in a 250 μm aperture in a Derlin cuvette using POPC, POPG, and cholesterol dissolved in *n*-decane at a 6:3:1 molar ratio. The buffer used was 1 M KCl + 10 mM HEPES, pH 7.4. Bilayer formation was followed by capacitance measurements using a 90-pF minimum threshold. The voltage clamp measurements were conducted using a high-gain electrophysiology amplifier with a resistive feedback headstage (Warner Instruments, BC-535), a digitizer (Digidata 1440 A), and a workstation computer. Data were recorded with a sampling frequency of 10 kHz. An integrated 8-pole low pass Bessel filter was used to filter the data at 1.0 kHz bandwidth. For further filtering and noise reduction, the data were also collected in parallel with another Bessel filter (Warner Instruments, LPF-8) at a cutoff frequency of 60 Hz. Data analysis was carried out with Clampfit (v10.6, Axon Instruments, San Jose, CA).

Peptide samples were prepared by dissolving in 1% $NH_4OH$ and diluting into a working buffer (1 M KCl +10 mM HEPES, pH 7.4). The peptide was added to one of the wells of the cuvette (trans side) at a final concentration of 100 nM, incubated for 5 min, followed by current recordings at various hold voltages, i.e., −100 mV, −50 mV, 50 mV, 100 mV (the sign corresponds to the trans side). Blockade of the channels by $Zn^{2+}$ ions was tested by adding $ZnCl_2$ to the trans side.

## Electrophysiology data analysis
The current traces were analyzed in a pClamp analysis environment. Between 120 s and 180 s snippets were selected, where the electrical activity was visually clear. Amplitude histograms of these snippets were generated to display the distribution of conductance values.

Channel diameter ($d$) was estimated using the formula[30,40]:

$$d = \frac{\rho g_{ch}}{2} \left( 1 + \sqrt{1 + \frac{16l}{\pi \rho g_{ch}}} \right) \qquad (1)$$

where $\rho$ is the resistivity of the salt solution inside the channel, $g_{ch}$ is the measured single channel conductance, and $l$ is the channel length. The length of a β-barrel-like channel was estimated based on the number of amino acid residues in the β-strand ($N_\beta$), the displacement along the β-strand axis per amino acid residue ($a = 3.48$ Å), and the β-strand tilt angle (β= 30-40 degrees relative to the membrane normal): $l = aN_\beta \cos\beta$. For the solution resistivity in the channel, two values have been used, i.e., the bulk valued and an empirically corrected, 5-fold higher value, as described in the Supplementary Methods file.

## AFM imaging and data analysis
For AFM imaging, the peptides were suspended in a buffer containing 300 mM KCl and 10 mM HEPES (pH 7.4) to 1 μM final concentration, vortexed for 30 s, deposited on freshly cleaved mica, incubated for ~10 min, washed 3 times with the same (imaging) buffer, and scanned. To prepare peptide/lipid samples for AFM imaging, the lipids were suspended in the imaging buffer supplemented with 3 mM $CaCl_2$ and bath-sonicated to obtain unilamellar vesicles. The peptide was added at 1:1000 or 1:500 peptide-to-lipid molar ratio followed by a ~1 min bath-sonication. The sample was drop-casted on the freshly cleaved mica surface and incubated for 20 min at room temperature, washed 3 times with the imaging buffer and the formed supported lipid bilayers were imaged in the same buffer. AFM imaging was performed on a Multimode AFM (Bruker) controlled by a Nanoscope V controller and the images were collected in PeakForce Tapping mode with SNL-10 cantilevers (Bruker). The images were analyzed using a Nanoscope Analysis v1.5 software or the SPM analysis package Gwyddion. All images were line flattened and color scale adjusted. For generating particle height histograms, the "Particle Analysis" feature in Nanoscope Analysis was used.

## CD and fluorescence

The lyophilized lipids and peptides were dissolved in chloroform and hexafluoroisopropanol (HFIP), respectively. The peptide solution was dried by desiccation followed by addition of the buffer and vortexing for 5 min. To prepare the proteoliposomes, the chloroform solution of the lipid was combined with the HFIP solution of the peptide at a 50:1 total lipid-to-peptide molar ratio, desiccated for 1 h, suspended in the buffer and vortexed for 5 min. When needed, the peptide or proteoliposome samples were extruded through 100 nm pore-size polycarbonate membranes using an Avanti Polar Lipids mini extruder. CD and fluorescence spectra were measured using a J-810 spectropolarimeter equipped with a fluorescence attachment and a temperature controller (Jasco, Tokyo, Japan) at 20 °C and 37 °C. Ten scans were averaged for the CD spectra and 3 scans for the fluorescence spectra.

## Attenuated total reflection Fourier transform infrared spectroscopy

For ART-FTIR experiments, the chloroform solution of the lipids was mixed with the HFIP solution of the peptide at a 50:1 total lipid-to-peptide molar ratio and spread over the surface of a germanium plate (5 cm × 2 cm × 0.1 cm, cut at the 2 cm edges at a 45 ° bevel angle). The sample was dried by 1-h desiccation and assembled in a flow-through ATR sample cell (Buck Scientific, East Norwalk, CT). The buffer (25 mM NaCl + 25 mM Na,K-phosphate in $D_2O$, pD 7.2) was injected into the cell to hydrate the peptide-lipid sample and FTIR spectra were recorded using a Vector-22 FTIR spectrometer (Bruker, Billerica, MA, USA) at two polarizations of the incident light, i.e., parallel and perpendicular to the plane of incidence. Transmission spectra of the blank buffer were measured separately and used as reference to obtain the respective absorbance spectra.

## ATR-FTIR data analysis

The secondary structure of the peptides was determined based on peak-fitting of the amide I FTIR spectra. First, the spectra measured at parallel and perpendicular polarizations of the incident light relative to the plane of incidence were used to obtain a "polarization-independent" spectrum, as described in the Supplementary Methods. Then, the areas of various amide I components were assigned to certain secondary structure types as follows: α-helix: 1660–1646 cm⁻¹, unordered: 1645–1638 cm⁻¹, β-sheet: 1637–1623 cm⁻¹, turn or other structures: 1705–1661 cm⁻¹. The areas of the components, corrected using the respective integrated molar absorptivities, relative to the total amide I area represented the fractions of various secondary structures.

The orientations of α-helical and β-sheet components were determined based on the dichroic ratios, $R = a_{\text{II}}/a_{\perp}$, where $a_{\text{II}}$ and $a_{\perp}$ are the amide I areas of a given structural component at parallel and perpendicular polarizations of the incident light, respectively. The order parameter of lipid acyl chains in supported membranes was determined based on the dichroic ratio of the methylene stretching vibrations in the spectral region 3015 cm⁻¹–2810 cm⁻¹.

## Reporting summary

Further information on research design is available in the Nature Portfolio Reporting Summary linked to this article.

## Data availability

Electrophysiological data for all four peptides at various applied voltages and respective conductance histograms, the effect of $Zn^{2+}$ ions on ion-conducting channels, light scattering, additional fluorescence, CD, and FTIR spectra as well as AFM images are provided in the Supplementary Information file. Any other data such as CD spectra of unextruded samples in the presence of lipid or thioflavin-T spectra, can be obtained from the authors upon request.

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

## Acknowledgements

Financial support from Florida Department of Health, Ed and Ethel Moore Alzheimer's Disease Research Program (grant 21A06) to S.A.T., UCSD MAE Departmental Fund and an anonymous donation to R.L. lab are gratefully acknowledged.

## Author contributions

S.A.T. and R.L designed the research; A.G.K, R.H., A.N., B.B., F.A. and S.A.T. conducted the experiments, A.G.K, R.H., R.L., and S.A.T. analyzed the data and prepared the figures, A.G.K, R.L., and S.A.T. wrote the manuscript.

## Competing interests

The authors declare no competing interests.
