## [Peer review file · Nature Communications]

REVIEWER COMMENTS

Reviewer #1 (Remarks to the Author):

This is an interesting study on the formation of oligomeric amyloid pores by 4 distinct neurotoxic forms of Alzheimer's β -amyloid peptide (A β). In contrast with the dominant but outdated paradigm that amyloid plaque formation is the root cause of Alzheimer's disease, the responsibility of amyloid channel formation in disrupting Ca²⁺ homeostasis is gaining more and more consideration. In this respect, this well-planned and well-done study warrants publication in Nature Communications.

Suggestions:

Lines 80-82. "These effects have been eliminated by peptides or small molecules that specifically block A β channels (refs 4,15-17) or prevent A β aggregation and membrane insertion (ref 21)."
It would be fair to also mention the following reference of a peptide inhibitor of A β membrane insertion and oligomerization (International journal of molecular sciences, 24(2), 1760.
<https://doi.org/10.3390/ijms24021760>).

Lines 232-233 . "In all cases except A β 1-40, the presence of 233 membranes appears to induce partial α -helical structure."
This appears to be in contradiction with the Abstract (lines 54-59) where α -helical structure is not mentioned. More generally, the authors should indicate the percentage of α -helical and β structures of the studied peptides in a lipid environment.

Lines 450-452. "The peptides have been reconstituted in lipid membranes composed of POPC, POPG, and cholesterol to mimic the fluidity, the anionic surface charge, and cholesterol content of neuronal membranes".
Here the authors should mention lipid rafts and gangliosides which are the natural anionic lipids of the outer leaflet of the plasma membrane of brain cells.

Reviewer #2 (Remarks to the Author):

- What are the noteworthy results?

This work concentrates on spectroscopic and electrophysiological measurements to compare 4 important A β peptides (observed in Alzheimer's patients brain: A β (1-40), A β (1-42) and 2 pyroglutamylated peptides by 3 a.acids shorter. Noteworthy, the basis of their neurotoxicity (which however s not measured in this work but rather other results are reviewed) is correlated to their structure and surface exposure of tyrosines.

- Will the work be of significance to the field and related fields?

Yes

How does it compare to the established literature? If the work is not original, please provide relevant references.

Separate studies on either electrophysiological features and cytotoxicity have been performed: Several papers by Viles and co-w., namely, Bode et al., J.Biol.Chem.292 (2017) reports comparative properties of A β (1-40) and A β (1-43), while pyroglutamylated peptides have been studied by Cherny and co-w., namely, Gunn et al., J.Biol.Chem.vol 291(2016)
Indeed, not all 4 peptides have been included in one study.

- Is the methodology sound? Does the work meet the expected standards in your field?

- Does the work support the conclusions and claims, or is additional evidence needed?

The work is done technically very well and supports conclusions about secondary structure, solvent exposure of Tyrosines and currents as observed by electrophysiology.

- Are there any flaws in the data analysis, interpretation and conclusions? - Do these prohibit publication or require revision?

There are no big flaws, only minor errors – pointed out below.

- Is there enough detail provided in the methods for the work to be reproduced?

Materials and Methods are given as Supplementary Materials, which is ok.

Minor recommendations:

line 273, should be Fig 2I not H, check.

In Discussion a more detailed analysis of the work by others, especially those cited above, would be recommended. Also, if no cell or neural culture toxicity measurements are done by the present authors (I understand that is beyond their physicochemical background), more detailed analysis of toxicity measured by others could be provided in Discussion. Is only A β (1-42) toxic or are indeed the two pyroglutamylated peptided "super-toxic". What is the explanation? Larger less controlled channels, producing larger sikes and Ca influx?

There is discrepancy about β -sheet content as reported in the literature, please include those studies, which claim that some pyroglutamylated variants have a larger proportion of β -sheet.

I feel the conclusions could be made more clearly and also the paragr.pointing out what still needs to be done could be enlarged. For example, recently appeared a review of structural and imaging studies on different channels by Viles (2023) Angewandte Ch.. Please include this reference and discuss, how amyloid pores (channels) could be detected and labelled specifically in extracted neuronal membranes or even in vivo, in animal models.

Reviewer #3 (Remarks to the Author):

The authors study the interesting topic of membrane interactions and associated properties of several amyloid-beta variants. The membrane pore formation experiments are highly relevant for the pathogenesis of Alzheimer's disease because they address one of the suggested mechanisms for the toxicity of these peptides. The authors correlate the pore forming properties with the beta-sheet content and the solvent exposure of the Tyr residue. A strength of the work is the inclusion of the pyroglutamylated forms, which are less studied, but occur in substantial amounts in the brains of patients. However, I have some doubts regarding the interpretation of the data as described in the specific comments below.

Samples

A description of the studied systems (membrane composition, concentrations, pH, etc.) should be provided at the beginning of the Results section and any changes to that preparation mentioned for each experiment.

Was the peptide TFA free?

Voltage clamp experiments

I personally find that Fig. S5 is more relevant and provides more mechanistic insight than Fig. 1. The authors may consider to exchange the two figures and combine their description. What does "More detailed analysis" in line 167 on page 4 mean, just an expansion of the horizontal time axis? Does this section refer to the same experiments as those shown in Fig. 1?

I am surprised by the long-term behavior on the seconds time scale shown in Fig. S5C. Is the initial phase with predominantly closed pore due to the addition of the peptide to the membrane (incorporation and pore formation)? Do the pores spontaneously switch back to predominantly closed on a longer time scale?

Please indicate zero current also in Fig. S5.

Regarding Fig. S6, I wonder how reproducible the observations are. The behavior described for the first Zn²⁺ addition in panel B starts already before Zn²⁺ addition, and can therefore not be its consequence. What happens when the same volume of water or buffer is added as a control?

CD

Fig. 2D: It is not clear to me how extrusion can take away predominantly unstructured peptides, which can be expected to be in monomeric form or within aggregates that are smaller than those with peptides in the beta-sheet conformation.

The near UV CD signal of predominantly the single Tyr is relatively strong compared to the far UV signal. Has this been observed before in the many published CD spectra of the amyloid-beta peptide or similar systems?

Fluorescence

The red shifted fluorescence signal is puzzling. The maximum of the Tyr emission spectrum is not much affected by the polarity of the solvent and is found at 303 nm for Tyr in water (Lakowicz' book). Thus solvent exposure cannot explain the band. Deprotonation as the cause is also unlikely because deprotonated Tyr fluoresces only weakly. I neither understand the explanation at the end of page 7. A higher sensitivity of S2 than of S1 to solvent polarity could explain different excitation spectra of exposed and buried Tyr, but not the red-shifted emission and not its absence upon excitation at 275 nm (which would excite the solvent insensitive S0->S1 transition). If the red shifted signal were caused by absorption of Tyr, its intensity would be strong upon excitation at 275 nm, which is not observed. If a red shifted emission were indicative of solvent exposure as stated at the beginning of page 8, it should have been observed many times before, for example for unstructured peptides or upon protein denaturation. I am not aware of such observations.

Therefore, the authors may want to check whether dityrosine formation can explain their spectra. Helpful would be to test longer excitation wavelengths as it can be assumed that the reported red shifted signals with 230-240 nm excitation stem from the S0->S2 excitation and that the S0->S1 excitation should occur above 275 nm.

Did the authors check for a contamination from the extrusion membrane (for example by using a buffer sample without peptide), which could be present in different amounts in the different experiments?

Fig. S7A is not very useful, since the different spectra cannot be distinguished easily. A stacked representation would be better.

ATR-FTIR

I do not see a brown curve and Fig. 3.

The band near 1665 cm⁻¹ occurs in a rather unspecific region. Therefore the assignment to alphaII helices is an option but not a definitive assignment.

There is a minor inconsistency in the analysis of the secondary structure content. The extinction coefficients used were determined in H₂O, but the experiments were done in D₂O.

Polarized ATR-FTIR

That parallel and perpendicular polarizations refer to the plane of incidence should be mentioned in the main text.

I am not an expert in polarized ATR-FTIR spectroscopy but would expect a stronger CH₂ signal for perpendicular than for parallel polarization in Fig. S13. When the acyl chains are perpendicular to the Ge surface, then the transition dipole moments of the CH₂ stretching vibrations are parallel to the surface and thus can be best excited by the perpendicularly polarized radiation (perpendicular to the plane of incidence but parallel to the surface).

An up to 800 nm thick multilayer cannot be expected to be well oriented throughout, which should affect the result of the orientation analysis for the secondary structure elements. Was this considered? In the bottom row of Fig. 3, the dichroitic ratio hardly ever follows the absorption of the component bands. For beta sheets and alpha helices with a well defined orientation, I would expect a maximum or

a dip in the dichroitic spectrum that follows the contours of the component band. Such a feature should be particularly obvious in panel C for the strong beta sheet bands seen in panels A and B but instead there is a nearly linear change of the dichroitic ratio in this spectral region. The R values in Table S2 all agree within the error limits and no conclusions should be drawn from the small differences.

Tables S1 and S2 contain too many irrelevant digits.

AFM

The tip diameter should be stated.

Do the top row images show one membrane patch per image?

The 3D close up views in Fig. 4 are not very telling as it is not clear to me what I am looking at. Only panel Aii seems to show a pore.

Reviewer #4 (Remarks to the Author):

The authors did a good job characterizing the four A β isoforms, namely, A β 1-42, A β 1-40, A β pE3-42. However, several major weaknesses are noted that must be addressed before further consideration.

1- I think the authors should perform extra experiments/analysis to show the I-V curve for at last A β 1-42 pore, to prove that different current levels are indeed representing the dynamics states of the pore and not due to a leaky artificial membrane.

2-the concentration of Zn is saturating 10mMm; I believe if Zn ions can block the open pore, then micromolar concentration should work.

I suggest also another experiment for the Zn effect; by adding EDTA after a few seconds of pore closure by Zn. EDTA should eliminate Zn ions and pores should restore their activity.

3-the authors didn't explain the mechanism or the model of pore forming. Current traces belong to toroidal pore or carpet model.

4-In Fig1A, authors claimed that the current trace is pore gating, which mean that current in this trace correspond to one single pore. I am wondering if the authors can determine the single channel conductance of this gating pore and then the pore size.

Minor

the authors completely ignore the seminal work of Natalia Carulla about the comparison of electrophysiology behavior differences between different A β length.

Response to Reviewers' Comments

REVIEWER #1

This is an interesting study on the formation of oligomeric amyloid pores by 4 distinct neurotoxic forms of Alzheimer's β -amyloid peptide ($A\beta$). In contrast with the dominant but outdated paradigm that amyloid plaque formation is the root cause of Alzheimer's disease, the responsibility of amyloid channel formation in disrupting Ca^{2+} homeostasis is gaining more and more consideration. In this respect, this well-planned and well-done study warrants publication in Nature Communications.

Response:

Thank you for your enthusiasm and constructive advice.

Remark #1:

Lines 80-82. "These effects have been eliminated by peptides or small molecules that specifically block $A\beta$ channels (refs 4,15-17) or prevent $A\beta$ aggregation and membrane insertion (ref 21)."

It would be fair to also mention the following reference of a peptide inhibitor of $A\beta$ membrane insertion and oligomerization (International journal of molecular sciences, 24(2), 1760. <https://doi.org/10.3390/ijms24021760>).

Response:

We have now added the suggested reference in the revised manuscript.

Remark #2:

Lines 232-233. "In all cases except $A\beta_{1-40}$, the presence of membranes appears to induce partial α -helical structure." This appears to be in contradiction with the Abstract (lines 54-59) where α -helical structure is not mentioned. More generally, the authors should indicate the percentage of α -helical and β structures of the studied peptides in a lipid environment.

Response:

We agree with the reviewer. Given the restricted number of words, the Abstract contains only the most salient findings. The alpha helix constitutes a small fraction of the peptides' secondary structure and as such we focused primarily on the major structural component, i.e., the beta sheet. The fractions of four distinct secondary structures of the peptides in lipid environment, derived from ATR-FTIR, are presented in Table 1. These are more reliable than the estimates derived from CD because in ATR-FTIR experiments the peptide is totally embedded in lipid multilayers (the peptide in the aqueous phase above the lipid layer does not contribute to the signal as the evanescent field does not reach that far) while in CD experiments the signal comes from both the membrane-bound and free peptides.

Remark #3:

Lines 450-452. "The peptides have been reconstituted in lipid membranes composed of POPC, POPG, and cholesterol to mimic the fluidity, the anionic surface charge, and cholesterol content of neuronal membranes".

Here the authors should mention lipid rafts and gangliosides which are the natural anionic lipids of the outer leaflet of the plasma membrane of brain cells.

Response:

We agree with the reviewer that gangliosides and lipid raft play a role in pore stability as well as activity in amyloid channels in plasma membrane, initially reported much earlier by Arispe and Kawahara and also summarized in a recent review by Viles (2023). Gangliosides are mainly present at the outer leaflet of the plasma membrane [Vasques JF et al., Neural Regen Res. 2023, 18(1):81-86. doi: 10.4103/1673-5374.343890]. The toxic Abeta pores, on the other hand, form by both extracellular and intracellular pools of Abeta. The latter interacts with the ER, mitochondrial and possibly other membranes which contain other types of anionic lipids (e.g. cardiolipin and PG in mitochondria). Therefore, we preferred to have a meaningful fraction of anionic lipid to mimic the general (rather than specific) anionic character of biomembranes. In order to have relevance of our work for both plasma membrane and intracellular membranes, we have used “neuronal membranes”, instead of “neuronal plasma membranes”.

REVIEWER #2

Remark #1:

- What are the noteworthy results?

This work concentrates on spectroscopic and electrophysiological measurements to compare 4 important A β peptides (observed in Alzheimer’s patients brain: A β (1-40), A β (1-42) and 2 pyroglutamylated peptides by 3 a.acids shorter. Noteworthy, the basis of their neurotoxicity (which however s not measured in this work but rather other results are reviewed) is correlated to their structure and surface exposure of tyrosines.

Response:

Thank you for highlighting the relevance of our work.

Remark #2

- Will the work be of significance to the field and related fields?

Yes

Response:

Thank you for the support and encouragement.

Remark #3:

How does it compare to the established literature? If the work is not original, please provide relevant references.

Separate studies on either electrophysiological features and cytotoxicity have been performed: Several papers by Viles and co-w., namely, Bode et al., J.Biol.Chem.292 (2017) reports comparative properties of A β (1-40) and A β (1-43), while pyroglutamylated peptides have been studied by Cherny and co-w., namely, Gunn et al., J.Biol.Chem.vol 291(2016) Indeed, not all 4 peptides have been included in one study.

Response:

We agree with the reviewer. We have several publications from our group as well as from others,

including those listed by the reviewer. We have limited the number of citations in this manuscript since it is not a review article, nevertheless, the majority of relevant studies from many groups are cited.

Remark #4

- Is the methodology sound? Does the work meet the expected standards in your field?

- Does the work support the conclusions and claims, or is additional evidence needed?

The work is done technically very well and supports conclusions about secondary structure, solvent exposure of Tyrosines and currents as observed by electrophysiology.

Response:

Thank you.

Remark #5:

- Are there any flaws in the data analysis, interpretation and conclusions? - Do these prohibit publication or require revision?

There are no big flaws, only minor errors – pointed out below.

Response:

Thank you.

Remark #6:

- Is there enough detail provided in the methods for the work to be reproduced?

Materials and Methods are given as Supplementary Materials, which is ok.

Response:

Thank you.

Recommendations:

Recommendation #1:

line 273, should be Fig 2I not H, check.

Response:

We have re-checked the text as well as Fig. 2 (Fig 3 in the revised manuscript). Data in Fig. 3I are with lipid, while the relevant main text is about data without lipid, i.e. Fig. 3H.

Recommendation #2:

In Discussion a more detailed analysis of the work by others, especially those cited above, would be recommended.

Response:

Bode, Baker and Viles (2017) paper was mentioned in the Introduction and the Discussion of the initial submission as Ref. 22. We have added some more details in the revised manuscript. For example, a new sentence is now added to the Discussion section “This is consistent with effective ion channel formation in membranes of HEK293 cells by A β ₁₋₄₂ but not A β ₁₋₄₀, which

was attributed to differences in membrane interaction and insertion abilities of the two A β species [Bode...2017].” Gunn et al. 2016 JBC paper had been cited in both the Introduction and the Discussion as well (original Ref. 6). Please see below for more information about this issue.

Recommendation #3:

Also, if no cell or neural culture toxicity measurements are done by the present authors (I understand that is beyond their physicochemical background), more detailed analysis of toxicity measured by others could be provided in Discussion. Is only A β (1-42) toxic or are indeed the two pyroglutamylated peptides “super-toxic”. What is the explanation? Larger less controlled channels, producing larger sikes and Ca influx?

Response:

Thank you for the excellent advice. We had discussed this issue briefly in the Introduction section of the original manuscript. Higher toxicity of pE-Abeta peptides vs the unmodified counterparts is well documented (e.g., Nussbaum et al., Nature. 2012, 485(7400):651-5. doi: 10.1038/nature11060). The hypertoxic nature of pE-Abeta has been indicated in the first paragraph of the Introduction of the initial submission. Regarding the mechanism of augmented cytotoxicity of pE-Abeta, conflicting data have been reported, such as faster or slower beta-sheet formation and aggregation. In this work, we focus on the membrane pore formation activities of the four Abeta peptides. In this context, the Discussion of the original submission had a sentence “The high frequency of opening of relatively large channels formed by A β pE₃₋₄₂ and A β pE₃₋₄₀ is likely to contribute to their augmented cytotoxicity, partially confirmed by induction of more efficient Ca²⁺ influx into neurons caused by A β pE₃₋₄₂ as compared to A β ₁₋₄₂⁶.” One additional sentence has now been added to the revised manuscript (page 14): “It is interesting to note in this context that A β pE₃₋₄₀ exerted stronger cytotoxic effect on cultured rat hippocampal neurons and cortical astrocytes than A β ₁₋₄₀, A β ₁₋₄₂, or A β pE₃₋₄₂ [Russo...2002], which might be related to its ability to form larger channels in cell membranes.”

Recommendation #4:

There is discrepancy about β -sheet content as reported in the literature, please include those studies, which claim that some pyroglutamylated variants have a larger proportion of β -sheet.

Response:

Two sentences are added in the first paragraph of the Introduction section of the revised manuscript: “The augmented cytotoxicity of A β pE has been attributed to elevated β -sheet formation and fibrillogenesis propensity [He and Barrow, 1999; Gunn...2016; Schilling..2006, Schlenzig 2009]. However, other publications have shown similar β -sheet structure and slower fibril formation by A β pE species as compared to their unmodified counterparts [Sanders...2009, Wirths...2010, Bouter...2013; Scheidt...2017].”

Recommendation #5:

I feel the conclusions could be made more clearly and also the paragr. pointing out what still needs to be done could be enlarged. For example, recently appeared a review of structural and imaging studies on different channels by Viles (2023) Angewandte Ch.. Please include this reference and discuss, how amyloid pores (channels) could be detected and labelled specifically in extracted neuronal membranes or even in vivo, in animal models.

Response:

The suggested paper (Viles, 2023) had already been cited in the initial submission as Ref. 24. It is now mentioned two times, once in the Introduction: “More details on membrane channel/pore formation by various A β species can be found in a recent review by Viles [Viles 2023]” and once in the Discussion: “...and high-resolution structures of A β membrane pores by methods such as NMR or cryo-electron microscopy or tomography [Viles 2023].”

REVIEWER #3

The authors study the interesting topic of membrane interactions and associated properties of several amyloid-beta variants. The membrane pore formation experiments are highly relevant for the pathogenesis of Alzheimer's disease because they address one of the suggested mechanisms for the toxicity of these peptides. The authors correlate the pore forming properties with the beta-sheet content and the solvent exposure of the Tyr residue. A strength of the work is the inclusion of the pyroglutamylated forms, which are less studied, but occur in substantial amounts in the brains of patients. However, I have some doubts regarding the interpretation of the data as described in the specific comments below.

Response:

Thank you for overall enthusiasm and for the constructive critiques and queries.

Remark #1

Samples:

A description of the studied systems (membrane composition, concentrations, pH, etc.) should be provided at the beginning of the Results section and any changes to that preparation mentioned for each experiment.

Response:

The buffer used is now identified in all Figure legends in addition to the SI Materials and Methods section.

Remark #2: Was the peptide TFA free?

Response:

The FTIR spectra indicate yes, the peptide samples are TFA-free. TFA generates a strong and sharp absorbance band at 1673 cm⁻¹, and this has not been observed in our experiments.

Remark #3:

Voltage clamp experiments::

I personally find that Fig. S5 is more relevant and provides more mechanistic insight than Fig. 1. The authors may consider to exchange the two figures and combine their description. What does "More detailed analysis" in line 167 on page 4 mean, just an expansion of the horizontal time axis?

Response:

“More detailed analysis” has been replaced with “A close-up view...” Figure S5 has been moved to the main text as Fig. 2.

Remark #4:

Does this section refer to the same experiments as those shown in Fig. 1?

I am surprised by the long-term behavior on the seconds time scale shown in Fig. S5C. Is the initial phase with predominantly closed pore due to the addition of the peptide to the membrane (incorporation and pore formation)? Do the pores spontaneously switch back to predominantly closed on a longer time scale?

Response:

During a voltage clamp experiment, after the addition of the peptide and stirring, there is usually a time lag between the addition of the peptide and manifestation of membrane electrical activity. Hence, the initial phase with predominantly closed state could potentially be indicative of peptide insertion and in-membrane pore structure formation. Over longer timescales, we have observed that the pores do spontaneously switch back to closed state, but also display interconversion to a different conductance state (Figure S3A and S3C). As described earlier by Arispe and Kagan labs independently, multiple pore-like species are reported to exist simultaneously in the membrane patch and each species can dynamically oligomerize and convert to a different conductance state.

Remark #5: Please indicate zero current also in Fig. S5.

Response:

We have made the change. Fig. S5 of the initial submission is now Fig. 2 in the main text.

Remark #6:

Regarding Fig. S6, I wonder how reproducible the observations are. The behavior described for the first Zn²⁺ addition in panel B starts already before Zn²⁺ addition and can therefore not be its consequence. What happens when the same volume of water or buffer is added as a control?

Response:

The blocking activity of Zn²⁺ ions is highly reproducible and has been documented extensively in the literature on amyloid ion channels.

In Fig S5 (Fig S6 in the original submission) before the first addition of Zinc, the current is seen in the form of frequent bursts. After the addition of Zinc, the frequency of current outbursts reduces. However, we noticed an unusual behavior where random, long duration of channel opening were observed.

It is likely that Zinc blocks the current elicited by oligomeric species which allow Zinc binding. Amyloid channels can adopt a conformation such that Zinc-binding amino acid Histidine at residue numbers 6, 13, 14 lines the pore opening. The sudden occurrence of a channel current after addition of Zinc could hint towards the presence of electrically active oligomeric species which do not adopt the conformation described above.

As suggested by the reviewer, we performed an experiment where we added Zinc (at similar concentrations) to membrane preparations without any amyloid. Voltage clamp steps according to our usual experimental protocol did not induce any membrane current (in the figure

below, current through the membrane is shown in the top panel (above red line) and the voltage waveform applied is shown below the red line). Hence the effect observed in Fig S5B could strongly be due to the complex polymorphic nature of A β .

Remark #7:

CD

Fig. 2D: It is not clear to me how extrusion can take away predominantly unstructured peptides, which can be expected to be in monomeric form or within aggregates that are smaller than those with peptides in the beta-sheet conformation.

Response:

Since Ab1-40 tends to be unordered more than the other three peptides, the presence of unordered large aggregates is possible. However, we have changed the sentence to a more conservative (and less speculative) version: "...and after extrusion the spectrum displayed one minimum at 217 nm assigned to β -sheet."

Remark #8:

The near UV CD signal of predominantly the single Tyr is relatively strong compared to the far UV signal. Has this been observed before in the many published CD spectra of the amyloid-beta peptide or similar systems?

Response:

In over 20 published papers that we accessed, CD spectra of Abeta peptides were presented in the 190 nm--260 nm region or narrower. In general, the L_b band of tyrosine side chain appears around 280 nm and has significant extinction coefficient ($1400 \text{ M}^{-1} \text{ cm}^{-1}$) which more than doubles ($2900 \text{ M}^{-1} \text{ cm}^{-1}$) upon deprotonation [Woody RW, Dunker AK. (1996) Aromatic and cysteine side-chain circular dichroism in proteins. In Circular Dichroism and the Conformational Analysis of Biomolecules, Fasman, G. D., ed, pp. 109–157, Plenum Press]. Figure 11 of that publication shows the spectrum of colicin A in 200 nm--320 nm region. The spectrum at pH 7 shows intensity at 222 nm of $\sim 21,000 \text{ deg cm}^2 \text{ dmol}^{-1}$ and intensity at 280 nm of $\sim 15,000 \text{ deg cm}^2 \text{ dmol}^{-1}$, indicating that the relative intensity of the aromatic side chains can be quite large.

In our work, the near-UV CD feature has not been used for drawing conclusions, hence it has not been further discussed in the revised manuscript.

Remark #9:

Fluorescence:

The red shifted fluorescence signal is puzzling. The maximum of the Tyr emission spectrum is not much affected by the polarity of the solvent and is found at 303 nm for Tyr in water (Lakowicz' book). Thus, solvent exposure cannot explain the band. Deprotonation as the cause is also unlikely because deprotonated Tyr fluoresces only weakly. I don't understand the explanation at the end of page 7.

Response:

There are two solvent effects, general and specific. The former is loss of part of the excited state energy due to solvent relaxation (or re-orientation) leading to red-shifted emission in more polar solvents. This effect is much less pronounced for Tyr than Trp, leading to an idea that Tyr emission is not solvent sensitive

We are interested in the specific solvent effect (page 7). A base, e.g., acetate or phosphate, forms H-bond with the phenolic hydroxyl and allows its deprotonation upon excitation (Lakowicz book, pp. 450-452 (1999 edition): "...the phenolic group of tyrosine can ionize even at neutral pH, and the extent to which this occurs depends on the base concentration and exposure of tyrosine to the aqueous phase."). So, in terms of specific solvent effect, observed in our work, Tyr emission does depend on solvent exposure. Regarding the relative emission intensities of protonated and deprotonated tyrosines, Munishkina and Fink (doi: 10.1016/j.bbamem.2007.03.015) present Tyr fluorescence spectra of alpha-synuclein at pH 7.5 and 10.5. The former displays emission peak around 307 nm and the latter shows a decrease in the 307 nm component and an additional emission band at 330-345 nm of equal intensity. Sethuraman and Rajendran (doi: 10.1021/acsomega.8b02928) also report Tyr emission peaks of equal intensity at 311 and 334 nm in protonated (pH 2.8) and deprotonated states (pH 8.43), respectively.

Remark #10:

A higher sensitivity of S2 than of S1 to solvent polarity could explain different excitation spectra of exposed and buried Tyr, but not the red-shifted emission and not its absence upon excitation at 275 nm (which would excite the solvent insensitive S0->S1 transition).

Response:

It is known that deprotonated Tyr emission is red-shifted to around 340 nm. At high pH, deprotonation occurs in the ground state. At neutral pH, deprotonation can occur in the excited state, especially when proton acceptors are present, including water. As the S0-S2 transition absorbs more energy than the S0-S1 transition, and in addition has a dipole along the hydroxyl group, excited-state deprotonation occurs during S0-S2 transition (at lower excitation wavelengths), facilitated by the solvent. The 275 nm excitation does not provide enough energy for deprotonation and red-shifted emission even in the presence of proton acceptors, hence it is solvent insensitive. See more below (item # 12).

Remark #11:

If the red shifted signal were caused by absorption of Tyr, its intensity would be strong upon excitation at 275 nm, which is not observed.

Response:

The red-shifted signal is present because of the S0-S2 excitation, which is solvent-sensitive. It cannot be stronger with S0-S1 excitation (at 275 nm) because it is absent (or negligible) at that excitation. See also answers to Remarks # 10 and # 12.

Remark #12:

If a red shifted emission were indicative of solvent exposure as stated at the beginning of page 8, it should have been observed many times before, for example for unstructured peptides or upon protein denaturation. I am not aware of such observations.

Response:

The red-shift effect of Trp fluorescence has been used for studying protein unfolding (e.g., doi: 10.1007/978-1-0716-2784-6_16). The paucity or absence of such studies using Tyr fluorescence may be due to the fact that (almost) all studies use Tyr excitation around 270-275 nm, which, as indicated above, excites the solvent-insensitive S0-S1 transition. It is possible that the emission from deprotonated tyrosinate at high pH (ground state deprotonation mechanism) can be detected with any excitation, including 275 nm, but detection of the solvent-exposed tyrosine (excited-state deprotonation mechanism, see Willis and Szabo 1991 J Phys Chem 95, 1585-1589) requires higher energy excitation to the S2 singlet state, as discussed above (item # 10).

Remark #13:

Therefore, the authors may want to check whether dityrosine formation can explain their spectra. Helpful would be to test longer excitation wavelengths as it can be assumed that the reported red shifted signals with 230-240 nm excitation stem from the S0->S2 excitation and that the S0->S1 excitation should occur above 275 nm.

Response:

Formation of dityrosine in our study is unlikely for several reasons: a) Dityrosine formation is achieved by hydrogen peroxide (doi:10.1371/journal.pone.0100200) and should not be formed spontaneously at any significant quantities, b) Its excitation and emission occur at higher wavelengths (315 nm and 400 nm, respectively), c) we have specifically checked this possibility; no fluorescence was detected with excitation around 315 nm.

Remark #14:

Did the authors check for contamination from the extrusion membrane (for example by using a buffer sample without peptide), which could be present in different amounts in the different experiments?

Response:

Could the red-shifted emission be due to contamination (such as tryptophan)? This is unlikely given the high purity of the peptide samples, the high intensity of the red-shifted component (often more than the basic Tyr fluorescence) and the fact that filtering (extrusion) increases the red-shifted signal (the final sample was always collected from the opposite side of the filter, see Materials and Methods: 15 passages).

Remark #15:

Fig. S7A is not very useful, since the different spectra cannot be distinguished easily. A stacked representation would be better.

Response:

We had considered the option of a stacked presentation. However, the main purpose of this Figure is to show the non-monotonous wavelength dependence of the scattered light intensity, and stacked presentation would be less effective. Usually, one would expect a decrease of scattering with increasing wavelength, whereas in fluorescence spectra we see the opposite, which will raise questions why. Figure S7 (now Figure S6) and its caption explain this effect, hence we would like to retain this Figure in its current form.

Remark #16: ATR-FTIR. I do not see a brown curve and Fig. 3.

Response:

The side chains contribute minimally to the amide I region. The brown component is seen in Fig. 3J (now Fig. 4J) (at the right edge) and a smaller one also in panel E. These components have not been used in calculations of the peptides' secondary structure.

Remark #17:

The band near 1665 cm⁻¹ occurs in a rather unspecific region. Therefore the assignment to alphaII helices is an option but not a definitive assignment.

Response:

Although the assignment of the component at 1665 cm⁻¹ to alpha(II) helix is reasonable, we agree that other structures, specifically type II beta-turns, could generate that signal as well (pp. 310-311 of Krimm S, Bandekar J. Adv Protein Chem. 1986;38:181-364. doi: 10.1016/s0065-3233(08)60528-8). We have added the following text to the revised ms. to address this issue: Given the spectral overlap of type II β -turn and α_{II} -helical structures⁵⁸, these ~4 amino acid residues may alternatively be assigned β -turn structure (page 11) and "although β -turn structure is also possible" (Table 1).

Remark #18:

There is a minor inconsistency in the analysis of the secondary structure content. The extinction coefficients used were determined in H₂O, but the experiments were done in D₂O.

Response:

If the extinction coefficients in D₂O (ϵ_D) and H₂O (ϵ_H) are related through an arbitrary coefficient k , ($\epsilon_D = k \cdot \epsilon_H$), in Eq. 2 of the SI the fraction of i th component in D₂O will be $f_i(D_2O) = k f_i(H_2O) / k = f_i(H_2O)$. This is because the fractions of secondary structures depend on the relative, not absolute, values of the extinction coefficients.

Remark #19:

Polarized ATR-FTIR

That parallel and perpendicular polarizations refer to the plane of incidence should be mentioned in the main text.

Response: We have included this clarification.

Remark #20:

I am not an expert in polarized ATR-FTIR spectroscopy but would expect a stronger CH₂ signal for perpendicular than for parallel polarization in Fig. S13. When the acyl chains are perpendicular to the Ge surface, then the transition dipole moments of the CH₂ stretching vibrations are parallel to the surface and thus can be best excited by the perpendicularly polarized radiation (perpendicular to the plane of incidence but parallel to the surface).

Response:

The x, y, and z components of the electric field of the evanescent wave have certain values, which depend, in particular, on the angle of incidence and the refractive indices. The Reviewer is correct assuming that for an ideally perfect membrane with all methylenes oriented perpendicular to the membrane normal the dichroic ratio (R) would be less than one. According to the theory, that value (for our case, i.e. 45 degrees incident angle....) is 0.854. As the order of the acyl chains decreases, R increases and assumes a value of 2.000 for a totally disordered sample. For more information, please see Tamm and Tatulian, Q Rev Biophys. 1997 30(4):365-429. doi: 10.1017/s0033583597003375 (871 citations).

Remark #21:

An up to 800 nm thick multilayer cannot be expected to be well oriented throughout, which should affect the result of the orientation analysis for the secondary structure elements. Was this considered?

Response:

Under our experimental conditions, the evanescent wave has a decay length of around 400 nm from the Ge surface and hence reports the order mainly within that region. Of course, the membrane is not expected to be homogeneously ordered throughout; we only measure the time- and ensemble-averaged order parameter. The “time- and ensemble-averaged” definition is added to the SI file (text above Eq. 3). Angular brackets are added in Eq. 3 and in Table S1. The word “average” and the angular brackets are added to the main text (p. 11).

Remark #22:

In the bottom row of Fig. 3, the dichroic ratio hardly ever follows the absorption of the component bands. For beta sheets and alpha helices with a well defined orientation, I would expect a maximum or a dip in the dichroic spectrum that follows the contours of the component band. Such a feature should be particularly obvious in panel C for the strong beta sheet bands seen in panels A, B but insteadof the dichroic ratio in this spectral region.

Response:

The feature observed by the reviewer reflects the significant spectral overlap of the amide I components of different secondary structures (alpha helix, irregular structure, beta sheet) as well as the less-than-perfect orientational order of the system. The dichroic ratios for beta sheet and alpha-helical structures have been determined using the areas of respective amide I components at two polarizations rather than the value of the dichroism at a certain wavenumber. This means

that the dichroic ratios and the derived order parameters are the time- and ensemble-averaged values, as already addressed above.

Remark #23:

The R values in Table S2 all agree within the error limits and no conclusions should be drawn from the small differences.

Response:

We have conducted *t* test analysis of the S_L values using Graphpad and have found the following, which has been added to the revised manuscript: “The difference between the effect of $A\beta_{1-42}$ and the other three peptides was statistically significant as judged from the *p* values when comparing S_L in the presence of $A\beta_{1-42}$ with those in the presence of $A\beta_{1-40}$, $A\beta pE_{3-42}$, and $A\beta pE_{3-40}$ (*p* = 0.0468, 0.0374, and 0.0089, respectively).” The following sentence has been added at the end of the SI file: “The statistical analysis of the S_L values was conducted using an online *t* test calculator (<https://www.graphpad.com/quickcalcs/ttest1/?format=SD>).”

Remark #24:

Tables S1 and S2 contain too many irrelevant digits.

Response:

We have corrected this issue. Now the numbers of decimals are more consistent.

Remark #25: AFM. The tip diameter should be stated.

Response:

SNL-10 tips (Bruker, CA) were used throughout the study which have a nominal tip diameter of 2nm. This has been added to the methods section in SI.

Remark #26:

Do the top row images show one membrane patch per image?

Response:

The top row AFM height images indicate the overall distribution of peptide oligomeric structures on one of the representative areas of the membrane patch. These areas were chosen mainly to display membrane homogeneity with a strong contrasting mica background. Multiple locations on the sample were selected and images were acquired in those areas.

Remark #27:

The 3D close up views in Fig. 4 are not very telling as it is not clear to me what I am looking at. Only panel Aii seems to show a pore.

Response:

We have added arrow markers to the image to guide the reader in the interpretation according to the text. We could resolve the pore-like structure only for $A\beta_{1-42}$. For $A\beta pE_{3-42}$, cluster of structures which resemble a channel-like structure were observed, but the internal pore could not be resolved. For $A\beta pE_{3-40}$, a weak, multi-unit peak can be observed, but again the internal pore

could not be resolved under the experimental conditions of AFM. For A β 1-40 however, we see only a single peak-like structure without any features. Significantly, without cholesterol in the bilayer, we have resolved pore-like features in these amyloid isomers in our published work.

REVIEWER #4

The authors did a good job characterizing the four A β isoforms, namely, A β 1-42, A β 1-40, A β pE3-42. However, several major weaknesses are noted that must be addressed before further consideration.

Response: Thank you.

Remark #1

1- I think the authors should perform extra experiments/analysis to show the I-V curve for at least A β 1-42 pore, to prove that different current levels are indeed representing the dynamics states of the pore and not due to a leaky artificial membrane.

Response:

Leaky membranes generally display baseline current drift, change in membrane capacitance over time (>10% from baseline capacitance). We did not observe these effects.

In control experiments with no peptide added, voltage steps did not cause any changes in the current, i.e., no sign of a leaky membranes. The figure below is an example of a voltage-clamp current recording: current trace (above the red line) with its corresponding applied voltage waveform (below the red line). Before the first addition of Zinc, the current trace is indistinguishable from the $I = 0$ pA baseline at 100 mV applied voltage confirming that no leakage currents exist.

We also performed tests to generate an I-V curve for A β 1-42 which shows an average conductance of 600-700 pS. This conductance is well within the values reported for amyloid channels. In previous work from ours and collaborators' labs (see references below the I-V Figure), we have reported similar conductances and behavior for A β 1-42 channels.

- [1] Connelly L, Jang H, Teran Arce F, Capone R, Kotler SA, Ramachandran S, Kagan BL, Nussinov R, Lal R. Atomic Force Microscopy and MD Simulations Reveal Pore-Like Structures of All-d-Enantiomer of Alzheimer's β -Amyloid Peptide: Relevance to the Ion Channel Mechanism of AD Pathology. *J Phys Chem B*. 2012 ,11,(5):1728-35 (Figure S3 and S4)
- [2] Hirakura Y, Lin M-C, Kagan BL. 1999. Alzheimer amyloid Ab31–42 channels: Effects of solvent, pH, and Congo Red. *J Neurosci Res* 57:458–466. (Fig 1C)
- [3] Arispe N, Pollard HB, Rojas E. Giant multilevel cation channels formed by Alzheimer-disease amyloid beta-protein [Ab-P(1–40)] in bilayer membranes. *Proc. Natl. Acad. Sci. USA* 90, 10573–10577 (1993).

Remark #2:

-the concentration of Zn is saturating 10 mM; I believe if Zn ions can block the open pore, then micromolar concentration should work.

Response:

In this work, we mainly focused on the structural and functional differences among the major isoforms A β 1-42, A β 1-40 and their pyroglutamylated variants. As Zn²⁺ has been conventionally shown to be an amyloid ion channel blocker, we wanted to verify if those characteristics are retained by the channel structures observed in this study. Hence, we worked with a saturating concentration of Zinc.

Arispe et al. [Arispe N, Pollard HB, Rojas E. *PNAS*. 93(4): 1710–1715 (1996)] in their seminal work showed that in the case of amyloid ion channels, a wide range of conductances can be observed due to the different aggregation states of A β and these electrically active oligomeric states can also interconvert among themselves to form channels with giant nS conductance. Arispe et al show that for channels exhibiting conductance <400pS, micromolar concentration of Zinc (200 μ M used) is sufficient to block the conductance (reversibly) and for channels

exhibiting conductance $>400\text{pS}$, millimolar concentration of Zinc was needed to block the conductance (2 mM used). In our experiments, we also observe channels covering a wide range of conductance (for 100 mV traces especially). Hence, we worked with 10 mM concentration of Zn^{2+}

Remark #2A:

I also suggest another experiment for the Zn effect; by adding EDTA after a few seconds of pore closure by Zn. EDTA should eliminate Zn ions and pores should restore their activity.

Response:

In the above-cited paper by Arispe et al., o-phenanthroline, a Zn^{2+} complexing agent is used to rescue conductance that was blocked by Zn^{2+} . It was shown that conductances $<400\text{pS}$ and $>400\text{pS}$ which were blocked by micromolar and millimolar concentration of Zn^{2+} ions, could be reversed by the addition of o-phenanthroline. Multiple other studies have also demonstrated similar effects. As this study concentrates more on the structural aspects that can influence channel forming propensities, a detailed Zn^{2+} scavenging and conductance reversal experiments were beyond the scope of the study.

Remark #3:

-the authors didn't explain the mechanism or the model of pore forming. Current traces belong to toroidal pore or carpet model.

Response:

This manuscript is not intended to address the issue of 3D conformations and modeling. We and others have published several papers on all four different amyloid beta isomers used in this study. In general, amyloid channels are generally heterodisperse in terms of their conductances, reflecting subunits heterogeneity and different channel structures formed by functionally different mechanisms. Our data suggest several scenarios:

$\text{A}\beta_{1-42}$ displays highest β -sheet content with respect to other peptides and also seems to display highly regular behavior in terms of current traces. As such, it should favor the most stable channels (the current trace is comparable to many standard ion channels). The calculated channel length (Table S3) in the case of $\text{A}\beta_{1-42}$ is 5.7 nm, which is fully membrane spanning and the order parameter calculated via polarized FTIR (Table S2) is the highest (higher even compared to plane lipid). This suggests that $\text{A}\beta_{1-42}$ channels adopt a highly stable configuration in the membrane and also stabilize the lipids surrounding them, which could hint at a membrane stabilized β -barrel pore. The increase in order parameter suggests cooperative interactions between the lipid and $\text{A}\beta_{1-42}$, leading to channels that persist for a long duration.

$\text{A}\beta_{\text{pE}3-42}$ exhibits the second highest fraction of β -sheet content among the 4 peptides. The channel length calculated is about 3.84 nm, i.e. long enough to form a transmembrane channel. The order parameter is very close to plane lipid, suggesting minimal perturbation of lipid arrangement and high degree of cooperative interactions between the peptide and the surrounding lipids.

$\text{A}\beta_{\text{pE}3-40}$ is ranked third in all the criteria discussed above with the length of the channel calculated to be 2.96 nm suggesting not fully transmembrane insertion. The nature of the current observed for $\text{A}\beta_{\text{pE}3-40}$ also displays very high variability and switching between levels indicating the 'unstable' nature of the channel. The order parameter also reduces in the case of $\text{A}\beta_{\text{pE}3-40}$, suggesting higher degree of perturbation to the lipid bilayer.

A β 1-40 ranks the lowest in all the categories with the lowest amount of β -sheet content and lowest degree of transmembrane insertion (2.0nm), suggesting it partially inserts into the upper leaflet of the bilayer displaying a dominant carpeting effect. The reduction in order parameter is almost comparable to A β pE3-40, suggesting higher degree of lipid perturbation and lower degree of cooperative interactions compared to A β 1-42 and A β pE3-42. The nature of current observed for A β 1-40 is highly unstable with mostly burst-like characteristics.

With these data and analyses, we can expect some oligomeric forms of A β 1-42 to adapt a β -barrel topology. A β pE3-42 and A β pE3-40 oligomers could be partially in β -barrel and toroidal pore topology. A β 1-40 data suggest a higher degree of carpeting effect.

The following text has been added to the revised manuscript: “Based on these results, A β 1-42 ($l = 5.7$ nm) is likely to form stable transmembrane channels, consistent with clear step-like current levels and increased lipid order (Table S2). A β pE3-42 may form transmembrane channels as well ($l = 3.84$ nm) with minimal effect on the lipid order, whereas A β pE3-40 ($l \approx 3$ nm) is likely to be incompletely membrane inserted, exerting a lipid destabilizing effect (Table S2). A β 1-40 ($l = 2$ nm) is not likely to form a transmembrane channel, consistent with the unique spiky current features generated by this peptide (Figs. 1, 2). Instead, it may cause membrane permeabilization by the “carpet” or detergent-like mechanisms. Still, A β 1-40 is included in Table S3 to show that if it were to form a β -barrel, then its pore diameter would vary between 0.34 and 0.66 nm, using the corrected solution resistivity.”

Remark #4

-In Fig1A, authors claimed that the current trace is pore gating, which mean that current in this trace correspond to one single pore. I am wondering if the authors can determine the single channel conductance of this gating pore and then the pore size.

Response:

Our data do not indicate voltage gating and we have not made such claims. In general, A β ion channels are not reported to exhibit gating behavior. In our experiments we have not found any voltage-dependent gating of channel dynamics with changing membrane hold voltage. Moreover, there is no known ligand-gating of A β ion channels as well.

Minor:

The authors completely ignore the seminal work of Natalia Carulla about the comparison of electrophysiology behavior differences between different A β length.

Response:

We believe that the two references already cited in the original manuscript represent major work from Dr. Carulla’s lab (original Refs. 28 and 42). We would have cited more publications if the current manuscript was a review article. Yet, if we have missed a specific and relevant paper, we are willing to consider adding it.

REVIEWER COMMENTS

Reviewer #3 (Remarks to the Author):

Thank you for the corrections and in particular for the explanation of the fluorescence results. Nevertheless, I have a few remaining issues:

1. Voltage clamp experiments (remark 6): The current trace in the provided figure shows a 60 s disturbance, when zinc is added. This is probably a normal artifact due to the injection of a solution. If that is true, it would be good to indicate the artifact signal in Fig S5 (with a bar or similar) because confusing a normal artifact with a real signal seems to have triggered my initial comment.

2. CD (remark 7): In the legend to the CD spectra in Fig. 3, the difference between full and dashed lines is not explained (extruded/not extruded?). Please add this information.

3. CD (remark 8): The near UV CD signal is still puzzling but I agree that it is hard to find spectra in the literature that cover the relevant range. However, the supplied reference does not help because there must be a mistake in the axis annotation: The larger of the two signals in panel b is $-15,000 \text{ deg cm}^2 \text{ dmol}^{-1}$ at 260 nm but it is zero at 250 nm in panel a. Extrapolating the slope near 260 nm in panel b indicates that the signal may reach zero at 240 nm. Thus there should be considerable ellipticity between 240 and 250 nm in panel a. I assume that the authors have run a buffer spectrum also. Is this horizontal (i.e. zero ellipticity) throughout the spectral range?

4. Fluorescence (remarks 9-13): Thank you for the extensive explanations. The Munishkina and Fink article states that also acidic protein residues can act as acceptors of the Tyr proton. If this is the case, then solvent access is not required, so the results can be explained with and without solvent access. I think it's worth to include some of the explanation in the main text, for example that the deprotonation is thought to take place from the S2 state. By the way, the spectra in the Sethuraman article were normalized, which explains the equal intensities at 311 and 334 nm.

5. ATR-FTIR (remark 18). There seems to be some confusion regarding the extinction coefficients at the end of page 18. The numbers are for the integrated absorption coefficients (B) but the units and the term "extinction coefficient" say otherwise. Some of the values come from ref 10 as stated, some from ref 9, but the source for the turn value is unclear (the sum of the values stated in ref 9?). This should be clarified. Regarding the absorption coefficients in H₂O and D₂O, one could indeed think that they are related by a uniform factor, but this is not supported by the literature. Chirgadze and coworkers (Biopolymers 12, 1337 and 13, 1701) have measured absorption coefficients and integrated absorption coefficients in D₂O. They get larger values for beta-sheets than for alpha helices, whereas the measurements by Venyaminov in H₂O indicate the opposite. I don't want to be picky and don't think that it matters too much but suggest to extend the description in the SI.

Reviewer #4 (Remarks to the Author):

I think that the authors have adequately addressed the comments made by the reviewers in the revised version of the manuscript. Therefore, I have no further comments.

REVIEWER COMMENTS

Reviewer #3 (Remarks to the Author):

Thank you for the corrections and in particular for the explanation of the fluorescence results. Nevertheless, I have a few remaining issues:

1. Voltage clamp experiments (remark 6): The current trace in the provided figure shows a 60 s disturbance, when zinc is added. This is probably a normal artifact due to the injection of a solution. If that is true, it would be good to indicate the artifact signal in Fig S5 (with a bar or similar) because confusing a normal artifact with a real signal seems to have triggered my initial comment.

Response: Thank you for pointing out this anomalous channel activity for A β 1-40. There are some scenarios where this 60 s conductance (disturbance) at ~110 s from the point of the addition of zinc could occur.

Arispe et al. [1] in their seminal work show that A β 1-40 channels that exhibit conductance states >400 pS need higher concentrations of Zinc to be blocked and even then, full blockage is not observed immediately. Along the same lines, we believe that the 60 s disturbance could be a high conductance state that was not immediately neutralized after the application of Zinc, but eventually dies out over the duration of the experiment.

Through ATR-FTIR experiments we show that A β 1-40 is the least membrane embedded among the 4 tested peptides. The unstable nature of the A β 1-40 current adds further evidence to this model. Due to the addition of Zinc, there could be disturbances at the peptide-membrane interface leading to the creation of metastable transitionary states which display the conductance behavior observed in the 60s disturbance. Over a longer time period, these states transition into permanently closed state as seen in the recording. Future work combining simultaneous structure-function studies will be needed to confirm this hypothesis.

Keeping Arispe et al. explanation and data from our ATR-FTIR experiments in mind, we would posit that the 60 s disturbance might not be an artifact, for the following reasons:

- a. In our study, under the same experimental conditions and protocols, recording from other peptides does not show such behavior. If it was indeed an artifact caused due to injection of solution, the artifact should have been present for all peptides.
- b. The 60 s disturbance happens after about 110 s following addition of zinc. A solution injection artifact could only have happened near the point of injection, thus confirming that it is not a solution injection artifact.
- c. The dynamics of the open state within the 60 s disturbance closely resemble the open state dynamics observable in ‘bursts’ of current observed for A β 1-40 in the same trace (burst events observable at ~55 s and ~85 s from the point of injection) and also in traces shown in Figs S2B and C (without the addition of zinc).
- d. In our first response to the reviewers, we have included a control current recording where zinc solution was added (twice) to a membrane patch that does not contain any peptides. No solution injection artifacts are observed in the control recording.
- e. In our current recordings of A β 1-40, 2 types of activity have been observed: - extremely

short-lived bursts of 15 ms-50 ms duration and long-lived bursts (open states >1 s). In the Zn blocking experiments for A β 1-40, it appears that after the first addition of Zn, short-lived burst currents are neutralized. The structures which elicit the long-lived bursts aren't fully neutralized (within a short period of time after zinc addition). It is likely that the 60 s disturbance could be due to these structures, which eventually do get neutralized over time (below we have included the full-length recording (at 100mV voltage clamp) which shows complete blockage, and no bursts are observed after the 60 s disturbance).

We have added the following two sentences to the revised main text: . This “anomalous” behavior of the current trace has only been seen for A β 1-40 but neither for the other three peptides nor for membranes without peptide addition. Hence, it reflects the unique structural and membrane interaction properties of A β 1-40, as discussed in the forthcoming sections.

[1] N Arispe, H B Pollard, E Rojas. Zn²⁺ interaction with Alzheimer amyloid beta protein calcium channels. Proc Natl Acad Sci U S A. 1996 Feb 20; 93(4): 1710–1715

2. CD (remark 7): In the legend to the CD spectra in Fig. 3, the difference between full and dashed lines is not explained (extruded/not extruded?). Please add this information.

Response: The legend to Figure 3 read “Dotted lines indicate that the samples have been extruded through 100 nm pore-size polycarbonate membranes.” In the revised text we have added “for both CD and fluorescence spectra.”

3. CD (remark 8): The near UV CD signal is still puzzling but I agree that it is hard to find spectra in the literature that cover the relevant range. However, the supplied reference does not help because there must be a mistake in the axis annotation: The larger of the two signals in panel b is -15,000 deg cm² dmol⁻¹ at 260 nm but it is zero at 250 nm in panel a. Extrapolating the slope near 260 nm in panel b indicates that the signal may reach zero at 240 nm. Thus there should be considerable ellipticity between 240 and 250 nm in panel a. I assume that the authors have run a buffer spectrum also. Is this horizontal (i.e. zero ellipticity) throughout the spectral range?

Response. Larger than expected intensity of near-UV CD could result from interactions of Tyr with Tyr or Phe side chains or main chain peptide bonds upon peptide aggregation. Such effects have been described by Strickland and Mercola (doi: 10.1021/bi00662a035) who report that near-UV CD intensities around 275 nm calculated for monomers, dimers, and hexamers of insulin were compatible with the experimentally observed CD spectra which were enhanced about fourfold in the hexamer compared with the monomer. Similar effects are likely to be involved in aggregation-prone peptides that contain both Tyr and Phe, such the A β peptides. However, we focus on the far-UV CD signal that can be interpreted in a straightforward manner

to probe the peptides' secondary structure, which is our primary purpose. No conclusions have been drawn based on near-UV CD as it would have been more speculative.

We have measured the CD spectra of the blank buffer and yes, it is a featureless horizontal line down to around 190-195 nm with increasing noise at lower wavelengths.

4. Fluorescence (remarks 9-13): Thank you for the extensive explanations. The Munishkina and Fink article states that also acidic protein residues can act as acceptors of the Tyr proton. If this is the case, then solvent access is not required, so the results can be explained with and without solvent access. I think it's worth to include some of the explanation in the main text, for example that the deprotonation is thought to take place from the S₂ state. By the way, the spectra in the Sethuraman article were normalized, which explains the equal intensities at 311 and 334 nm.

Response: To further clarify the processes behind fluorescence data, a new sentence has been added to the 2nd paragraph of the section Structural Features from Fluorescence Spectroscopy: "Thus, the whole process where deprotonation is involved is thought to proceed as follows: S₀→S₂ transition, excited state deprotonation facilitated by a proton acceptor, S₂→S₁ internal conversion, vibrational decay to the lowest energy level of S₁, emission, i.e. radiative transition to a vibrational level of S₀." In the same section, changes have been made to incorporate the following two sentences: "Tyr emission splitting may be facilitated by the solvent but also by amino acid side chains with proton acceptor properties such as aspartate or glutamate.⁵⁶ In the latter case, the effect would be unaltered upon change of the buffer." The last sentence of the paragraph has been modified: "While the involvement of amino acids with proton acceptor groups cannot be ruled out, clear differences between phosphate and Tris buffers and the unbuffered solution ~~These data~~ suggest that the splitting of Tyr fluorescence is caused by the solvent and is stronger when strong H-bonding acceptors and donors are present in the buffer.

5. ATR-FTIR (remark 18). There seems to be some confusion regarding the extinction coefficients at the end of page 18. The numbers are for the integrated absorption coefficients (B) but the units and the term "extinction coefficient" say otherwise. Some of the values come from ref 10 as stated, some from ref 9, but the source for the turn value is unclear (the sum of the values stated in ref 9?). This should be clarified. Regarding the absorption coefficients in H₂O and D₂O, one could indeed think that they are related by a uniform factor, but this is not supported by the literature. Chirgadze and coworkers (Biopolymers 12, 1337 and 13, 1701) have measured absorption coefficients and integrated absorption coefficients in D₂O. They get larger values for beta-sheets than for alpha helices, whereas the measurements by Venyaminov in H₂O indicate the opposite. I don't want to be picky and don't think that it matters too much but suggest to extend the description in the SI.

Response: The term "extinction coefficient" is changed to "integrated molar absorptivity" and ϵ is replaced with B. It's the same as B in Chirgadze et al. (1973, 1974) and Venyaminov and Kalnin (1990) papers. They use a unit of L mol⁻¹ cm⁻². When liter (L) is replaced with 1000 cm³, we have 1 L mol⁻¹ cm⁻² = 1000 cm³/mol, so B = 7.6×10⁴ L mol⁻¹ cm⁻² for α -helix (Venyaminov and Kalnin 1990) becomes 7.6×10⁷ cm³/mol.

Finding precise values of B is tricky as reported data are scarce and inconsistent. Chirgadze et al. (1973, 1974) report values for deuterated proteins. Full deuteration was achieved either by extensive dialysis against D₂O (1.5 days, with 10 replacements of the D₂O bath) or by

heating (60 degrees C) in D₂O for 12 hours. Our peptide samples were lipid-embedded, followed by addition of D₂O-based buffer. It is known that the membrane-embedded segments of proteins that are not directly exposed to the solvent undergo slow HD exchange (time constants 2-3 hours, doi: 10.1016/s0014-5793(98)00091-x), so the spectral measurements (1-1.5 hours) were completed by the time when less than 50% of the peptide was deuterated implying that the use of B values for undeuterated peptides is justified. FTIR measurements on partially deuterated proteins is a common practice and the best way to distinguish between α -helix and unordered structures (doi: 10.1007/978-1-4939-9512-7_13), again justifying the experimental and data analysis procedures employed in this work. Hence, we found the values reported by Vanyaminov and Kalnin (1990) most useful. Values of B reported by these latter authors vary in the sequence $B_{\alpha} > B_{\beta} > B_{\text{unordered}}$. Values reported by Chirgadze et al. (1973, 1974) for deuterated proteins are smaller by 20-70% and the sequence is $B_{\beta} > B_{\alpha} > B_{\text{unordered}}$. Venyaminov and Kalnin knew this discrepancy as they cited both Chirgadze papers (and actually were working in the same lab). So, the statement of Venyaminov and Kalnin that the B values for random coil and beta conformation approximately coincide (page 1271) and that “The integral intensity of the α -helix in H₂O solution is equal to that of the β -conformation, but in D₂O is equal to that of the random coil” should be taken as an update of the data reported by Chirgadze et al. 16 years earlier.

As for the turn structure, in the revised manuscript we attribute that component to “other” structures, which may involve mainly various types of turns and other conformations for which an average value of $B_{\text{other}} = 5.5 \times 10^7$ cm/mol has been used. These amendments are presented by text on the revised SI file.

In conclusion, we would like to thank this reviewer for taking the time to carefully read our manuscript and to provide most useful and constructive criticism.

Reviewer #4 (Remarks to the Author):

I think that the authors have adequately addressed the comments made by the reviewers in the revised version of the manuscript. Therefore, I have no further comments.

Response: Thank you for reading our work and providing constructive comments.